

# Tropospheric CO vertical profiles measured by IAGOS aircraft in 2002-2017 and the role of biomass burning

Hervé Petetin[1], Bastien Sauvage[1], Mark Parrington[2], Hannah Clark[3], Alain Fontaine[1], Gilles Athier[1], Romain Blot[1], Damien Boulanger[4], Jean-Marc Cousin[1], Philippe Nédélec[1], Valérie Thouret[1]

[1]Laboratoire d'Aérologie, Université de Toulouse, CNRS, UPS, Toulouse, France
[2]European Centre for Medium-Range Weather Forecasts (ECMWF), Reading, United Kingdom
[3]IAGOS-AISBL, Brussels, Belgium
[4]Observatoire Midi-Pyrénées, Université de Toulouse, CNRS, UPS, Toulouse, France

*Correspondence to:* H. Petetin (hervepetetin@gmail.com)

**Abstract.** This study investigates the role of biomass burning and long-range transport in the anomalies of carbon monoxide (CO) regularly observed along the tropospheric vertical profiles measured in the framework of IAGOS. Considering the high interannual variability of biomass burning emissions and the episodic nature of pollution long-range transport, one strength of this study is the amount of data taken into account, namely 30,000 vertical profiles at 9 clusters of airports in Europe, North America, Asia, India and southern Africa over the period 2002-2017.

As a preliminary, a brief overview of the spatio-temporal variability, latitudinal distribution, interannual variability and trends of biomass burning CO emissions from 14 regions is provided. The distribution of CO mixing ratios at different levels of the troposphere is also provided based on the entire IAGOS database (125 million CO observations).

This study focuses on the free troposphere (altitudes above 2 km) where the long-range transport of pollution is favoured. Anomalies at a given airport cluster are here defined as departures from the local seasonally-averaged climatological vertical profile. The intensity of these anomalies varies significantly depending on the airport, with maximum (minimum) CO anomalies of 110-150 (48) ppbv in Asia (Europe). Looking at the seasonal variation of the frequency of occurrence, the 25% strongest CO anomalies appears reasonably well distributed along the year, in contrast to the 5% or 1% strongest anomalies that exhibit a strong seasonality with for instance more frequent anomalies during summertime in northern United-States, during winter/spring in Japan, during spring in South-east China, during the non-monsoon seasons in south-east Asia and south India, and during summer/fall at Windhoek, Namibia. Depending on the location, these strong anomalies are observed in different parts of the free troposphere.

In order to investigate the role of biomass burning emissions in these anomalies, we used the SOFT-IO v1.0 IAGOS added-value products that consist of FLEXPART 20-days backward simulations along all IAGOS aircraft trajectories, coupled with anthropogenic (MACCity) and biomass burning (GFAS) CO emission inventories and vertical injections. SOFT-IO estimates the contribution (in ppbv) of the recent (less than 20 days) primary worldwide CO emissions, tagged per source region. Biomass burning emissions are found to play an important role in the strongest CO anomalies observed at most airport clusters. The regional tags indicate a large contribution from boreal regions at airport clusters in Europe and North America during summer season. In both Japan and south India, the anthropogenic emissions dominate all along the year, except for the strongest summertime anomalies observed in Japan that are due to Siberian fires. The strongest CO anomalies at airport clusters located in south-east Asia are induced by fires burning during spring in south-east Asia and during fall in equatorial Asia. In southern Africa, the Windhoek airport was mainly impacted by fires in southern hemisphere Africa and South America.

To our knowledge, no other studies have used such a large dataset of in situ vertical profiles for deriving a climatology of the impact of biomass burning versus anthropogenic emissions on the strongest CO anomalies observed in the



troposphere, in combination with information on the source regions. This study therefore provides both qualitative and quantitative information for interpreting the highly variable CO vertical distribution in several regions of interest.

## 1  Introduction

Biomass burning represents a major source of pollution throughout the troposphere, with strong impacts on the atmospheric composition (Duncan et al., 2003; Hodzic et al., 2007; Sauvage et al., 2007; Konovalov et al., 2011; Parrington et al., 2012; Yamasoe et al., 2015), air quality (Bravo et al., 2002; Sapkota et al., 2005; Bowman and Johnston, 2005; Viswanathan et al., 2006) and radiative balance (Forster et al., 2007; Spracklen et al., 2008; Stone et al., 2008; Péré et al., 2014). Biomass burning here denominates both prescribed and natural open fires of vegetation
(savannah, forest, agricultural residues) and peat, thus excluding domestic biofuel combustion for cooking and heating (Langmann et al., 2009). Among the myriad of compounds emitted by these fires — aerosols (e.g. organic carbon, black carbon, inorganics), greenhouse gases (e.g. $CO_2$, $CH_4$, $N_2O$) and photochemically reactive gases (CO, $NO_x$, non-methane volatile organic carbon) — carbon monoxide (CO) represents the dominant species after carbon dioxide ($CO_2$) (Urbanski et al., 2008). Global CO vegetation fire emissions are estimated at about 433 TgCO year$^{-1}$ on average over
the 1997-2004 period (van der Werf et al., 2006), thus comparable with anthropogenic emissions that range between 476 and 611 TgCO year$^{-1}$ in 2000 depending on the inventory (Lamarque et al., 2010; Granier et al., 2011). Due to a long lifetime of around 1-3 months, CO plumes are subject to long-range transport from the regional to the hemispheric scale, as shown by a wide literature (e.g. Forster et al., 2001; Damoah et al., 2004; Colarco et al., 2004; Nédélec et al., 2005; Kasischke et al., 2005; Real et al., 2007; Stohl et al., 2006, 2007). In the boreal regions, in contrast with most
anthropogenic emissions primarily confined to the planetary boundary layer (PBL), compounds emitted during open fires may be subject to pyro-convection, allowing a quick uplift in the free troposphere (Val Martin et al., 2010) and even the lower stratosphere under extreme conditions (Fromm et al., 2000; Fromm and Servranckx, 2003; Jost et al., 2004; Fromm et al., 2005; Trentmann et al., 2006; Cammas et al., 2009). At such altitudes, long-range transport is again favoured by stronger winds, sometimes allowing plumes to circumnavigate the world in 2-3 weeks (Damoah et al.,
2004; Dirksen et al., 2009).

Our understanding of the impact of the biomass burning remains limited by the numerous uncertainties on emissions, plume transport and chemical evolution. Despite persistent uncertainties, satellite observations have allowed major progresses in characterizing the spatial and temporal distribution of biomass burning emissions (see Langmann et al. (2009) for an overview of burned area and active fire satellite products). However, due the wide variety of parameters
involved in such combustion processes — e.g. fuel content, combustion completeness, burning conditions (flaming, smouldering or both) — and the subsequent high variability of emissions depending on the geographical region (Urbanski et al., 2008), characterizing the chemical composition of vegetation fires plumes and its evolution remains challenging. In terms of transport, main uncertainties concern the injection height that depends in a complex way on the released fire energy and meteorological conditions (e.g. wind speed, stability, water vapour) (Freitas et al., 2007;
Langmann et al., 2009; Val Martin et al., 2010).

Although satellite observations can provide valuable information on the impact of biomass burning, they remain limited by their coarse vertical resolution. Assessing the large-scale impact of biomass burning plumes therefore requires airborne observations in the free troposphere where the transport of plumes is favoured. During the last decades, many airborne campaigns have been designed to shed light on vegetation fires, e.g. YAK-AEROSIB (Airborne Extensive
Regional Observations in SIBeria) (Paris et al., 2008), POLARCAT (Polar Study using Aircraft, Remote sensing, surface measurements and models, of Climate, chemistry, Aerosols and Transport) (Pommier et al., 2010) or BORTAS (BOReal forest fires on Tropospheric oxidants over the Atlantic using Aircraft and Satellites) (Palmer et al., 2013).



These campaigns have provided detailed information on fire plumes but remain somewhat limited by their short time coverage. Frequent profiles with high vertical resolution are essential for better characterizing biomass burning plumes and their transport. In the framework of the MOZAIC program and its successor the IAGOS European Research Infrastructure (ERI) (the MOZAIC-IAGOS programs are hereafter denoted IAGOS), a large dataset of $O_3$ and CO

vertical profiles (obtained during ascent and descent phases) is available for many parts of the world since 1994 and 2002, respectively. Of the more than 300 airports served by IAGOS aircraft, several have been sufficiently visited over the period 2002-2017 to establish reliable climatological vertical profiles based on which anomalies can be discriminated on a daily basis. This study provides an overview of the CO anomalies observed in the vertical profiles over the period 2002-2017 (distribution, height, seasonal variations), and investigates the influence of vegetation fires

versus anthropogenic emissions on these anomalies as well as the source regions. One major current deficiency of Eulerian models is their inability to resolve persistent (vertically) thin plumes due to a rapid dissipation by numerical diffusion in sheared flows (Eastham and Jacob, 2017), due to too coarse vertical resolution in the free troposphere (Zhuang et al., 2018). Thus, this study addresses this problem by using the SOFT-IO tool (Sauvage et al., 2017b) that couples FLEXPART Lagrangian backward simulations with CO emission inventories. Although some results will be

shown in the tropics, this study will mainly focus on the northern mid-latitudes where most IAGOS profiles are available. In total, about 30,000 CO profiles are included in this analysis. To our knowledge, this is the first study that addresses the question of the biomass burning impact on tropospheric CO based on such a large dataset of in-situ measurements and over such a long period (16 years).

The input data and the modelling tools used in this study are described in Sect. 2. A description of the CO vegetation

fire emissions over the period 2002-2017 is provided in Sect. 3. An overview of the tropospheric CO profiles is given in Sect. 4 while the analysis of the CO anomalies is presented in Sect. 5. Results are discussed in Sect. 6.

## 2   Material and methods

### 2.1   IAGOS observations

This study mostly relies on the CO observations available in the framework of the IAGOS ERI ([www.iagos.org](www.iagos.org))

(Petzold et al., 2015). Observations are performed by commercial aircraft from several airline companies since 1994 for ozone and 2002 for CO. In both the MOZAIC and IAGOS programs, the same instruments are used in all aircraft. During the 2011-2014 overlapping years, inter-comparisons have been systematically performed between MOZAIC and IAGOS, demonstrating a good consistency in the dataset (Nédélec et al., 2015). In MOZAIC, ozone was measured using a dual-beam UV-absorption monitor (time resolution of 4 seconds) with an accuracy estimated at about ±2 ppbv /

±2% (Thouret et al., 1998), while CO was measured by an improved infrared filter correlation instrument (time resolution of 30 seconds) with a precision estimated at ±5 ppbv / ±5% (Nédélec et al., 2003). In IAGOS, both compounds are measured with instruments based on the same technology used for MOZAIC, with the same estimated accuracy and the same data quality control. A more detailed description of the IAGOS system and its validation can be found in Nédélec et al. (2015).

Of the 300 or so airports visited for two decades, this study focuses on those with sufficient observations to build reliable seasonally-averaged climatological vertical profiles. In order to increase the amount of available data and fill data gaps, airports less than 500 km apart are combined into airport clusters following the description given in Table 1. The location of these airports is shown in Fig. 1. We consider only the profiles available in a validated status (i.e. after post-flight calibration) in the IAGOS database over the period 2002-2017. The total number of profiles (with at least

one IAGOS CO measurement) is 29,904 over that period. Note that although most IAGOS profiles of 2017 are not yet



fully validated, we still include this year in the analysis because many profiles at ChinaSE (and few profiles at AsiaSE) are already calibrated and validated in 2017.

### 2.2    IAGOS data treatment

For convenience in the data treatment and the presentation of results, all IAGOS profiles are first averaged over 250 m thick layers from 0 to 12.5 km above sea level (ASL) (i.e. values given at 125 m include observations between 0 and 250 m). This study focuses on the troposphere. For all profiles, the tropopause altitude is identified based on the potential vorticity (PV) fields extracted from ECMWF (European Center for Medium-range Weather Forecast) operational analysis (00:00, 06:00, 12:00, 18:00 UTC) and forecasts (03:00, 09:00, 15:00, 21:00 UTC) and interpolated on a 1°x1° global longitude-latitude grid. The tropopause is located at the pressure level where PV reaches the threshold of 2 pvu (potential vorticity unit). Thus, stratospheric intrusions in the troposphere are discarded. Similarly to Petetin et al. (2015, 2016b), only the part of the profiles within a radius of 400 km around the airport is retained. This ensures that we do not take into account the cruise phase of the flight, especially in the tropics where aircraft never reach the (much higher) tropopause.

### 2.3    Source apportionment with the SOFT-IO tool

In order to get information about the recent contributions of the different CO emission sources, we use the recently developed SOFT-IO v1.0 tool (Sauvage et al., 2017b, 2017a). Here we only give a brief overview of SOFT-IO; more details can be found in the reference paper of Sauvage et al. (2017b). The SOFT-IO data are freely available in the IAGOS database (www.iagos.org) (https://doi.org/10.25326/3, Sauvage et al., 2017a).

Along all aircraft trajectories, SOFT-IO couples FLEXPART retro-plume simulations over 20 days with anthropogenic and biomass burning CO emission inventories. At any given point in the IAGOS trajectories, it thus provides an estimate of the primary CO contribution (in ppbv) of the recent (20 days or less) worldwide emissions. Anthropogenic and biomass burning contributions are computed separately in order to discriminate between both origins. Additionally, the contributions are quantified for the 14 different source regions defined in GFED emissions (see Fig. 1). Among the different emission inventories available, we will use in this study the monthly MACCity anthropogenic emissions (Diehl et al., 2012; Lamarque et al., 2010; Granier et al., 2011; van der Werf et al., 2006) and the daily GFAS biomass burning emissions (Kaiser et al., 2012). As GFASv1.2 is not available in 2002, we use GFASv1.0 for this first year and GFAS will hereafter denominate the combination of GFASv1.0 in 2002 and GFASv1.2 from 2003 to 2017. Note that both inventories agree well over the overlap period 2003-2010. In SOFT-IO, the MACCity and GFAS inventories are considered at a longitude-latitude resolution of 0.5x0.5° and 0.1x0.1°, respectively. Anthropogenic emissions are applied in the first layer above ground (0-1 km). However, vegetation fires are usually associated to fast updraft, including pyro-convection, and their emissions thus need to be injected in altitude. Various vertical distributions of fire emissions have been proposed in the literature but are still affected by major uncertainties (Val Martin et al., 2010). Among the several approaches available in SOFT-IO, we use in this study the injection height recently provided by ECMWF, based on the fire observations and operational weather forecasts of ECMWF (Paugam et al., 2015; Rémy et al., 2017). As this last product is not available during 2002, we use the MIXED injection profiles during this year. The MIXED injection profiles consist in a combination of injection profiles of Dentener et al. (2006) in the tropics and mid-latitudes, and injection profiles deduced from a look-up table computed with the plume rise model PRMv2 of (Paugam et al., 2015) (see Sauvage et al., 2017b for more details).

SOFT-IO does not calculate the CO background; this unaccounted background here represents the primary CO from emissions older that 20 days and secondary CO (oxidation of $CH_4$ and non-methanic volatile organic compounds).





### 3 Description of the CO biomass burning emissions

Before investigating the CO anomalies (Sect. 4) and the role of biomass burning emission sources (Sect. 5), we provide in this section a brief overview of several general aspects of the GFAS biomass burning emissions over the period 2002-2017, namely their spatio-temporal variability (Sect. 3.1), latitudinal distribution (Sect. 3.2) and seasonal trends (Sect.

3.3). This will help the interpretation of the results in the following sections. In order to avoid confusion, all seasons hereafter will be given in their boreal sense : winter for December-January-February (DJF), spring for March-April-May (MAM), summer for June-July-August (JJA) and fall for September-October-November (SON).

### 3.1 Spatio-temporal variability

The seasonal biomass burning CO emissions from the GFAS inventory are plotted in Fig. 2 at the global scale and for

the different regions (see Fig. S1-2 in the Supplement for a similar plot of anthropogenic and total CO emissions). The mean CO emissions and their inter-annual variability (IAV, here calculated as the standard deviation normalized by the mean) in the different regions are reported in Table 2. The acronyms of the different regions are also indicated. Note that these regional emission estimates are in general agreement with those given by Kaiser et al. (2012) with the GFASv1.0 over the period 2003-2011 (although the definition of the regions slightly differs).

On average over the period 2002-2017, the global biomass burning emissions are 361 TgCO yr$^{-1}$. This represents 38% of the total (anthropogenic plus biomass burning) CO emissions when considering the MACCity anthropogenic emission inventory. Emissions from biomass burning mostly come from continental tropical regions (SHAF, NHAF, SHSA) and BOAS. The other regions of interest are EQAS, SEAS, AUST and BONA. At the global scale, annual fire emissions have a relatively low IAV of 13%. This is notably due to the high contribution of African fires (35% of the

global fire emissions) that have the lowest IAVs among all 14 regions, below 11%. In contrast, vegetation fire emissions strongly vary in most of the other regions. The highest IAV is observed in EQAS (90%) due to the well-known influence of the El Niño Southern Oscillation (ENSO) (van der Werf et al., 2008). This is illustrated by the very strong emissions that occurred in late 2015 concomitantly with a strong ENSO (Yin et al., 2016; Lohberger et al., 2018). In this region, the IAV during the fire season (SON) reaches 128% (the highest seasonal IAV among all regions

and seasons). The IAV in BOAS is 49% at the annual scale and about 70% in spring/summer. In BONA, the other region of interest in northern extra-tropics, the variability is relatively lower, 38% at the annual scale and 48% in summer. Note that when considering total CO emissions, the highest IAVs are found in EQAS (58%), BOAS (41%) and AUST (38%).

In boreal regions, several factors drive the intensity of biomass burning emissions, including weather, carbon fuel

content and topography. In particular, the spatiotemporal variability of fire emissions can be linked to the presence of persistent high pressure systems (i.e. anticyclones) in which dry air masses remain confined. This is illustrated in Fig. S3 in the Supplement by the summertime geopotential height anomalies at 500 hPa (Z500) (i.e. the height of the 500 hPa pressure surface above mean sea-level) given by the ERA-interim reanalysis relatively to the 1974-2017 climatology. High values of Z500 correspond to anticyclonic conditions, thus favourable to fires. In BOAS, the intense

emissions in summer 2003 and 2012 (40 and 66 TgCO yr$^{-1}$, respectively) were observed in the regions of strong positive anomalies of Z500 in central Siberia. Similarly, strong (peat) fires were observed in 2010 around Moscow (Konovalov et al., 2011) concomitantly to a high Z500 anomaly (above 10 decametres); however, at the scale of the whole region, CO emissions remain close to their average. In BONA, a Z500 anomaly of similar magnitude was observed in summer 2004 in Alaska, again associated to major fires (26 TgCO yr$^{-1}$) (Turquety et al., 2007; Pfister et al.,



2006). Similar emissions are observed in summer 2013-2015 but with much lower Z500 anomalies than in 2004, which illustrates the influence of the abovementioned other factors.

### 3.2 Latitudinal distribution

In order to highlight how the respective contributions of anthropogenic and biomass burning emissions vary depending on the latitude and the season, the latitudinal distribution of CO emissions from these two sources is shown in Fig. 3. The anthropogenic CO emissions peak in the 20°N-40°N band during all seasons. Conversely, the latitudinal distribution of biomass burning emissions strongly varies with the season. In winter (DJF), they are maximum in the 0°N-10°N band due to fires in NHAF. In spring (MAM), their distribution shows two modes, in the 0-30°N and 50°N-60°N bands mostly due to fires in SEAS and BOAS, respectively. In summer (JJA), two clear modes also appear with strongest emissions in the 20°S-0 and 50°N-70°N bands due to fires in southern hemisphere (SHAF and SHSA) and boreal regions (BOAS and BONA), respectively. In fall (SON), the distribution highlights only one main mode in the 20°S-0 band mainly due to biomass burning in southern hemisphere (SHAF, SHSA and AUST) and EQAS. The strongest relative contributions of biomass burning to total emissions are found (i) in the southern hemisphere during all seasons at a varying distance from the equator depending on the position of the inter-tropical convergence zone (ITCZ) and (ii) at high latitudes in the northern hemisphere during all seasons except in winter.

### 3.3 Seasonal trends

In this section, we investigate briefly the trends of CO biomass burning emissions given at the seasonal and regional scale by the GFAS inventory over the period 2002-2017. Considering the potentially strong IAV of fires (Sect. 3.1), it is worth keeping in mind that such a 16 year-long period may still be too short to give robust trend results.

We calculated the linear trends of CO emissions over the period 2002-2017 for all seasons and regions (Table 3). All trends uncertainties are given at a 95% confidence level. At the global scale, the GFAS inventory depicts a significant decrease of CO emissions of -1.7±1.0% yr$^{-1}$, (-6.1±3.7 TgCO yr$^{-1}$) mostly due to decreasing emissions in winter and fall. Several regions show significant trends during specific seasons, although many of them correspond to a very low extra amount of CO released in the atmosphere. The most noticeable and strongest annual trend (-5.1±3.8% yr$^{-1}$ or -2.6±2.0 TgCO yr$^{-1}$) is observed in SHSA where CO emissions are decreasing during all seasons except winter (mostly in summer and fall). Using multiple satellite-derived fire products, Chen et al. (2013) investigated in detail the IAV and trends of fires in South America over the period 2001-2012. In particular, they highlighted an increase of the number of active fires over 2001-2005 followed by a slight decrease (and large IAV), notably due to a substantial reduction of deforestation in Brazil over the 2000s (Reddington et al., 2015). This is consistent with the GFAS emissions shown here. Extending the period of study to 2017 shows that CO emissions remained in the range of (relatively) low values over the last years, which explains the negative trends obtained here.

A substantial decrease of CO biomass burning emissions (-2.1±1.2% yr$^{-1}$ or -0.8±0.5 TgCO yr$^{-1}$) is also observed in NHAF during the fire season (winter). In CEAS, a significant decrease is found at the annual scale (partly driven by a decrease in fall). A strong but weakly significant decrease is also observed during summertime in EQAS (-6.5±6.1% yr$^{-1}$ or -0.6±0.5 TgCO yr$^{-1}$). Due to surprisingly higher emissions in 2017 (a factor 2-3 higher than over the period 2002-2016), the MIDE shows significant positive trends during all seasons but CO emissions in this region are very low. The strong emissions in 2017 are probably artificially caused by an out-of-date mask for filtering of oil and gas flaring hotspots in the GFAS system, which would not cover the more recent activities in this region. In most other regions, no significant trends are found.



## 4 Overview of the CO vertical profiles measured by IAGOS aircraft

### 4.1 Distribution of the CO mixing ratios in the entire IAGOS database

An overview of the distribution of all CO mixing ratios measured by IAGOS aircraft during the period 2002-2017 is shown in Figure 4 (see Fig. S4-7 in the Supplement for the seasonal distributions). Specifically in this section, the entire IAGOS dataset is taken into account (not only the tropospheric profiles available at the clusters of airports introduced in Sect. 2.1) in order to give the largest view on the CO mixing ratios measured in the atmosphere since the beginning of the MOZAIC program. Note that these distributions are not calculated based on the mean mixing ratios over 250 m thick layers but on the individual measurements. The statistical robustness is ensured by the large number of observations that reaches about 125 million. About 110 million (88%) are performed between 9 and 13 km (during the cruise phase of IAGOS aircraft), while the number of measurements elsewhere (ascent and descent phases) ranges between 1.1 and 2.5 million per 1-km layer. At the annual scale, the mean CO mixing ratios decrease from 210 ppbv at 0-1 km to 72 ppbv at 12-13 km. The $1^{st}$ percentile of the distribution decreases from 92 to 28 ppbv. The $99^{th}$ ($99.9^{th}$) percentile ranges from 750 (1,619) to 162 (229) ppbv. Above 9 km of altitude, the highest CO mixing ratios in the whole IAGOS database reach about 1,100 ppbv and was measured during summertime.

### 4.2 Climatological CO vertical profiles

The mean seasonal CO vertical profiles at the different airport clusters are shown in Fig. 5. As already described in Petetin et al. (2016), the CO mixing ratios at Germany airports decrease from 230 ppbv at the surface to 90 ppbv at 12 km (given that the stratosphere is filtered out). Close to the surface, much stronger CO mixing ratios (300-400 ppbv) are observed in Asia (strongest CO at AsiaSE, followed by Japan and ChinaSE). Compared to Germany, this corresponds to a relative difference between +20 and +60% (up to +80% for AsiaSE during winter). The lowest surface CO mixing ratios (about 100 ppbv) are measured at Windhoek, Namibia (elevation of 1,600 m), due to the fact that the airport is located at about 40 km from the city and is surrounded mostly by desert. Apart from Windhoek, slightly weaker differences between the airport clusters are found higher in altitude, usually between ±20%. One noticeable exception is Japan where CO mixing ratios above 8 km are 10-30 ppbv stronger than at the other clusters during spring/summer.

### 4.3 Individual CO vertical profiles

In this section, we give a brief overview of all CO vertical profiles measured at the different airport clusters. Although the question of the type (anthropogenic versus biomass burning) and geographical origin of the CO anomalies is addressed in Sect. 5, some first interpretations of the strongest plumes with SOFT-IO are provided here.

#### 4.3.1 Germany

The CO vertical profiles measured above German airports are shown in Fig. 6 for all years since 2002 (one panel per year). The profile availability throughout the year is indicated in blue. As the most frequently visited by the IAGOS fleet, the German airport cluster is particularly useful for monitoring the IAV of CO plumes sampled by aircraft. Both the number and the intensity of CO plumes strongly vary from one year to another. The strongest CO mixing ratios are observed in the free troposphere in 2003, 2005, 2012-2015. In 2005, these plumes are observed in the lower free troposphere during wintertime (the seasonal versions of Fig. 6 are not shown). According to SOFT-IO, they are mainly due to anthropogenic emissions from Europe. During the other years, the high CO mixing ratios (from 250 up to 500 ppbv) are measured higher in altitude (around 6-10 km) and mostly during summertime. These values greatly exceed the $99.9^{th}$ percentile of all IAGOS CO mixing ratios in this range of altitude (Sect. 4.1). Most of these years were associated with a strong fire activity in the boreal regions (Sect. 3.1), which suggests a noticeable contribution of biomass burning to these strong CO anomalies in Europe. In 2003, SOFT-IO also indicates that some pollution plumes





are due to European fires. Tressol et al. (2008) already analysed the influence of the intense Portugal fires on the IAGOS profiles during the heat wave of 2003. Between 2006 and 2011, CO mixing ratios roughly remain in the range of climatological values.

### 4.3.2    North America

The CO profiles measured at USeast are shown in Fig. 7. Very high CO mixing ratios exceeding 300 ppbv are observed in 2002 and 2015, as well as in 2003-2004 and 2011 although more sporadically. All of them are measured in summer, usually at higher altitudes than in Germany (above 4 km and up to 11 km). The number and intensity of CO plumes is highly variable from one year to the other. A part of this IAV is obviously due to the number of available profiles that ranges in summer from 31 in 2007 to 117 in 2013 and 2015 if we exclude the years without any data. However, these

differences of sampling are not likely to explain all of the variability. This can be illustrated by the summer 2013 during which only one high CO plume is observed despite the availability of 109 vertical profiles distributed almost every day of the season while much more layers are observed during several summers with sparser data (e.g., in 2014). In comparison, the CO plumes observed at USlake are less numerous, less intense and located at a lower altitude (see Fig. S8 in the Supplement). The strongest CO plumes at USlake are observed in 2015 with mixing ratios of 300-400 ppbv

between 2 and 5 km. At CAwest, profiles are much sparser, in particular during summer. Some strong CO enhancements are still observed, with mixing ratios reaching about 300 ppbv during spring 2006 and fall 2004, 2009, 2012 and 2015 (see Fig. S9 in the Supplement). Again, the altitude of these plumes is highly variable (from 3 to 12 km). Asian CO plumes of 200-300 ppbv were observed in North-eastern Pacific at altitudes between 3 and 9 km during the TRACE-P campaign (Heald et al., 2003).

### 20    4.3.3    Asia and India

The Fig. 8 shows the CO vertical profiles at the Japan cluster. Strongly polluted CO plumes are observed very frequently at this cluster, with mixing ratios exceeding 300 ppbv (up to 600 ppbv) between 2 and 12 km almost every year with a sufficiently large number of profiles. They are the most frequent and strongest in winter and spring (not shown). Some plumes are also observed during summer (mainly in 2002, 2003, 2005, 2012 and 2013). During fall, CO

plumes are preferentially observed at the beginning of the period between 2002 and 2004.

IAGOS profiles at ChinaSE airport cluster are less numerous and more irregularly distributed anomalies over the period 2002-2017 (Fig. 9). Pollution plumes are frequently observed through the entire free troposphere, with CO mixing ratios often exceeding 400 (300) ppbv below (above) 5 km. Most of these events occur during spring. Quite similar patterns are observed at the AsiaSE cluster, although CO anomalies are usually weaker, in particular in spring (see Fig.

S10 in the Supplement). At both airport clusters, numerous strong CO plumes are intercepted by IAGOS aircraft throughout the entire troposphere during fall 2015, with mixing ratios reaching 300 (500) ppbv at ChinaSE (AsiaSE). This intense pollution is likely due to the intense fires that burnt over Indonesia during the strong ENSO event in fall 2015 (Yin et al., 2016; Lohberger et al., 2018) (sect. 3.1). Note that no such strong mixing ratios are observed by IAGOS aircraft in fall during the other years with available measurements, as illustrated in Fig. S11 in the Supplement

by the comparison between fall 2015 and 2016, the two years with higher sampling frequency at these airport clusters (about 200 and 70 profiles per fall season at ChinaSE and AsiaSE, respectively). According to the CAMS interim reanalysis, the strong positive anomaly of the CO global burden caused by these fires during fall 2015 persisted into early 2016 (Flemming and Inness, 2017). This would be consistent with the IAGOS observations in south-east Asia that also exhibit relatively strong CO mixing ratios during winter and spring 2016 (not shown).



Among all airport clusters considered in this study, SouthIndia includes the lowest number of profiles (1,114). Most CO profiles at SouthIndia airport cluster are actually available in 2012-2014 (Fig. 10). In this region, spring (MAM) corresponds to the pre-monsoon period, summer (roughly JJA, up to September actually) to the monsoon period, fall (mostly October-November actually) to the post-monsoon period. Sheel et al. (2014) already investigated the CO

vertical distribution at Hyderabad based on some MOZAIC measurements. Close to the surface, CO mixing ratios are strongest in winter due to higher CO emissions (notably from coal and wood burning), more stagnant weather conditions and more generally a continental influence (Sheel et al., 2014; Verma et al., 2017). Conversely, the lowest CO is observed during the monsoon due to clean marine air masses (from Arabian Sea and Indian Ocean) brought by strong south-westerly winds (Sheel et al., 2014; Verma et al., 2017). A similar seasonality at the surface is observed at

different locations in India (Verma et al., 2017). In comparison, the seasonal variability is smoother in the free troposphere. Some moderately polluted plumes with mixing ratios up to about 300 ppbv are sampled below 4 km. Compared to the previous airport clusters, much fewer strong CO anomalies are observed high in the troposphere, with CO mixing ratios usually remaining below 200 ppbv (except for one profile in November 2015 in which a plume of 250 ppbv was observed at 9 km). Similarly to the surface, the lowest CO mixing ratios in altitude are found during the

monsoon season.

### 4.3.4   Windhoek (Namibia)

The CO profiles at Windhoek are shown in Fig. 11. Due to the remote location of Windhoek airport, the CO mixing ratios remain very low (80-100 ppbv) and nearly constant with altitude during winter and spring. All the strong CO anomalies are observed in summer and fall, which corresponds to the fire season in southern Africa and South America

(Sauvage et al., 2005). The strongest CO mixing ratios can reach 600 ppbv and are observed mostly in the lower troposphere (below 4 km ASL or 2.3 km AGL (above ground level)) but also higher in the troposphere (around 8 km, in 2007 for instance). Such high CO mixing ratios exceeding 400 ppbv in the lower troposphere are not observed every year (e.g. lower CO plumes in 2006 and 2011).

## 5   Analysis of the CO anomalies and contribution of vegetation fires

### 5.1   Methodology

As discussed in Sect. 4, the high IAV of the occurrence of strong CO anomalies and their usual coincidence with high fire activity in some nearby and/or upwind regions suggest a noticeable role of biomass burning sources. In this section, this role is investigated more quantitatively at the different airport clusters. For each 250 m thick altitude layer of each profile, we define the CO anomaly as the observed mixing ratio minus its corresponding seasonal climatological vertical

profile (calculated over the 2002-2017 period). Therefore, these CO anomalies can be positive or negative. This approach is chosen for its objectivity and simplicity. In this paper, since we are more interested in the long-range transport that is favoured in the free troposphere, only the anomalies above 2 km AGL are considered. In addition, this study will focus on the strongest positive CO anomalies. Different thresholds, $p$, expressed as a percentile of the CO anomalies distribution will be discussed and, for clarity, the corresponding subset will be annotated $CO^{>p}$. For instance,

$CO^{>75}$ and $CO^{>90}$ represent the 25% and 10% highest CO anomalies among the whole database, respectively (and thus $CO^{>0}$ represents the whole anomalies dataset). Note that all 250m-width layers are treated independently from each other. This means for instance that on a given profile, one single large pollution plume observed between 5 and 6 km of altitude will be treated as 4 (250m thick) anomalies.



For each CO anomaly, both biomass burning ($C_{BB}$) and anthropogenic ($C_{AN}$) contributions (in ppbv) are calculated with the SOFT-IO tool (Sect. 2.3). The $C_{BB}/C_{AN}$ ratio (unitless) is then used to characterize the predominant origin of the anomaly. CO anomalies with $C_{BB}/C_{AN}$ ratios above 2 (below 0.5) are considered as mainly influenced by BB (AN) emissions and are hereafter called BB-like (AN-like) anomalies. CO anomalies with intermediate $C_{BB}/C_{AN}$ ratios

between 0.5 and 2 are considered as a relatively balanced mix of BB and AN emissions, and are hereafter referred as MIX-like anomalies. It is worth keeping in mind that BB-like anomalies still include a contribution from AN emissions, and vice versa.

Some examples of vertical profiles at New York airport are given in Fig. 12. On the first profile on August 28th 2004, anomalies between 4-6 km (below 3 km and above 6.5 km) will be tagged as BB-like (AN-like). As previously

explained (Sect. 2.3), SOFT-IO does not simulate the CO background that represents in this example about 100 ppbv. The second example (Fig. 12, right panel) is shown in order to illustrate the uncertainties affecting the transport of the plume (leading in this case to a 1-km error in the altitude of the plume).

### 5.2    SOFT-IO contributions

The SOFT-IO tool was evaluated in Sauvage et al. (2017b) over the entire IAGOS dataset. Evaluation results have

shown that SOFT-IO detects more than 95% of all observed CO plumes. The biases in the CO enhancements are usually lower than 10-15 ppbv in most regions, although the agreement is lower in the middle troposphere possibly due to numerous thin plumes of low intensity (Sauvage et al., 2017b). Note that, as previously explained in Sect. 5.1, the way we define the CO anomalies in our study (departure from the seasonal climatological profile) differs from Sauvage et al. (2017b) (departure from a linear fit of the CO vertical profile above 2 km, plus additional conditions on the excess

of CO; more details can be found in Sect. 3.4 of Sauvage et al. (2017b)). Sauvage et al. (2017b) also reported stronger biases on the extreme plume enhancements. Several sources of uncertainty can explain the discrepancies, including the parameterization of the FLEXPART model, the meteorological fields, the emission inventories and, specifically for the biomass burning, the injection height. Nevertheless, SOFT-IO is meant to be a useful tool (especially in a qualitative perspective, but also quantitatively) for interpreting the CO mixing ratios measured by IAGOS aircraft.

In our study, we are not trying to quantify exactly the $C_{AN}$ and $C_{BB}$ contributions along all profiles. Instead, we are more interested in identifying the predominant type of emission sources (AN-like, BB-like or MIX-like) of all anomalies. In order to investigate how SOFT-IO performs in our methodology, we computed the distribution of simulated total ($C_{AN+BB}$=$C_{AN}$+$C_{BB}$) contributions over different 10 ppbv-wide bins of observed CO anomalies (Fig. 13). The distributions (box-a-whisker plots) are calculated only when the number of points in the bin exceeds 20.

Results at all airport clusters exhibit a general increase of the mean contribution simulated by SOFT-IO from the lowest (negative) to the highest (positive) observed CO anomalies. Note that we do not expect these plots to follow the 1:1 line since contributions and anomalies are not defined in the same way and are thus not directly comparable. However, this increase tends to flatten in the range of higher anomalies. This is consistent with the stronger negative biases reported by Sauvage et al. (2017b) for the CO plumes of strongest intensity. At some airport clusters (Germany, ChinaSE), both

the mean contribution and the strongest percentiles show a slight decrease in the highest anomalies. Reasons for this are not clearly identified. Due to a low number of points (below 100) in this range of extreme values, these distributions may not be as representative as for the anomalies of lower intensity. Nevertheless, these comparisons give us confidence on the ability of SOFT-IO to provide useful information regarding the CO anomalies observed in the IAGOS database, especially on the climatological point of view.



### 5.3 Seasonal distribution of the CO anomalies and influence of biomass burning

The seasonal distribution of CO anomalies is shown in Fig. 14 for the $CO^{>0}$ (i.e. all points), $CO^{>75}$, $CO^{>95}$ and $CO^{>99}$ subsets. The percentiles of the CO mixing ratios anomalies vary strongly depending on the airport (Table 4). For instance, the $CO^{>99}$ anomalies subset at Germany (Japan) include all points with CO mixing ratios at least 48 (151) ppbv

higher than the climatological value at the corresponding altitude and for the corresponding season. Among all airport clusters, Germany exhibits the lowest departures from the seasonally-averaged climatological vertical profiles. Japan shows the strongest 99th percentile of CO anomalies, followed by ChinaSE and AsiaSE. It remains high at Windhoek (96 ppbv), but much lower values are found at clusters located in North America and India (60-70 ppbv).

In order to make seasonal results comparable, as the number of flights varies depending on the season, all frequencies of

occurrence are weighted by the total number of available data during each season (which explains why all bars in the $CO^{>0}$ dataset are at 25%). The relative proportion of AN-like, MIX-like and BB-like anomalies at the seasonal scale is indicated on each bar.

Considering all the CO anomalies no matter their intensity (i.e. the $CO^{>0}$ anomalies set), results clearly indicate a dominant influence of the anthropogenic emissions whatever the season and the airport cluster. The only exception is

Windhoek where the proportion of MIX-like and BB-like anomalies is large, in particular during fall. This is roughly consistent with the fact that most airports are located in a latitudinal band where AN emissions have a dominant contribution to the total emissions, except for Windhoek (Sect. 3.2). The lowest contributions of AN-like anomalies are found in spring and/or summer at northern mid-latitudes and in winter at SouthIndia.

At all locations, the $CO^{>75}$ anomalies occur quite regularly all along the year. They remain dominated by anthropogenic

emissions except for Windhoek. However, results at mid-latitudes airports show that the slightly lower anthropogenic emissions in summer (and to a lesser extent in spring and autumn) are compensated by higher fire emissions that increase the frequency of occurrence of MIX-like and BB-like anomalies.

Considering the $CO^{>95}$ subset, some seasonal differences appear at most airport clusters. The most obvious seasonal pattern is observed at Windhoek (ChinaSE) where more than 60% (50%) of the anomalies occur during fall (spring). At

both locations, these anomalies appear a strong contribution of fires (still mixed with anthropogenic emissions at ChinaSE). At AsiaSE and SouthIndia, a much lower number of anomalies is found during summer. On airports on both sides of the Atlantic, anomalies are substantially less frequent in fall than during the other seasons. In particular, at USlake, more than 40% of these strong anomalies are concentrated in summer, with a substantial contribution of fire emissions. Located downwind of China, Japan show more frequent $CO^{>95}$ anomalies in winter and spring, essentially

due to anthropogenic emissions.

Looking at the 1% strongest anomalies ($CO^{>99}$ subset), results exhibit a quite similar picture although with exacerbated seasonal differences and stronger $C_{BB}$ contributions (except in Japan and SouthIndia for which AN emissions remain dominant). In particular, the frequency of occurrence of CO anomalies during spring in ChinaSE reaches 80% (and more than 70% for Windhoek in fall). However, it is worth noting that for this anomalies subset, caution is required at

all locations except Germany since the number of points is greatly reduced (between 300 and 600 points depending on the airports).

Therefore, the two main conclusions of this analysis are (i) the large seasonal variability of the CO anomalies with the strongest intensity in the free troposphere, and (ii) the growing influence of biomass burning sources (relatively to anthropogenic sources) as one looks at the strongest anomalies at all airport clusters except Japan and SouthIndia.

### 5.4 Vertical distribution of the CO anomalies

We now investigate where in the troposphere these CO anomalies are the more frequent. The frequency of occurrence of the CO anomalies is shown in Fig. 15 for the $CO^{>75}$, $CO^{>95}$ and $CO^{>99}$ subsets. For a given threshold, season and





altitude, the frequency is here calculated as the number of CO anomalies exceeding the threshold normalized by the total number of points available at this altitude during all seasons. As in Sect. 5.3, an adjustment factor is applied to balance the differences of sampling between the seasons. Note that as the total number of available points decreases at the highest altitudes (above 10 km), the results in this region of the troposphere are less robust than at lower altitudes.

At the annual scale, the $CO^{>75}$ anomalies are equally distributed in the free troposphere at most airport clusters, although some weak variations are observed in Asia (less frequent anomalies in middle troposphere) and at Windhoek (more frequent anomalies higher in the troposphere). Low to moderate differences are observed at the seasonal scale. Larger differences are found for the $CO^{>95}$ and $CO^{>99}$ subsets. At the Germany cluster, the strongest anomalies tend to be more frequent in the lower part of the free troposphere, except in spring and summer when anomalies are found higher in

altitude. At USeast and USlake, the strongest anomalies are more equally distributed in the troposphere although the frequency of occurrence drops above 10-11 km. Different results are observed at CAwest where the strong anomalies are the most frequent above 4-5 km in winter, spring and fall while there are also observed in the lower troposphere in summer. At Japan airports, frequent strong anomalies are observed in the upper troposphere (above 10 km) in spring. At ChinaSE and AsiaSE, the strongest anomalies are clearly the most frequent in the lower free troposphere in spring and

to a lesser extent in winter, and extend higher in the troposphere during fall. At SouthIndia, frequent anomalies are also in the lower free troposphere during all non-monsoon seasons, with a secondary maximum of frequency in the upper troposphere. At Windhoek, the strongest anomalies are restricted to the lower free troposphere during the burning season, except during fall when frequent strong anomalies are also observed higher in altitude, up to 10-11 km.

### 5.5    Geographical origin of biomass burning contributions

In this section, the geographical origin of the $C_{BB}$ and $C_{AN}$ contributions is investigated for the different CO anomalies subsets. Note that we are here no longer considering the different types of anomalies (AN-like, MIX-like, BB-like). Instead, we are analysing the mean $C_{BB}$ and $C_{AN}$ contributions for the different anomalies subsets and source regions.

#### 5.5.1    Germany

Fig. 16 shows the mean $C_{BB}$ and $C_{AN}$ contributions for different CO anomalies subsets at the Germany cluster, with

information about the geographical origin of the corresponding primary emissions. The proportion of the contribution in the total contribution ($C_{BB}/(C_{AN+BB})*100\%$) is also indicated with pie charts. The overall ($CO^{>0}$) mean total contribution is 12 ppbv, with seasonal averages ranging between about 10 ppbv in fall/winter and 14 ppbv in spring/summer (column of percentile 0 in the 5 panels of Fig. 16). Considering only the $CO^{>99}$ anomalies subset, the mean contribution reaches 35 ppbv at the annual scale, with seasonal values around 50 ppbv in summer and 25-30 ppbv

during the other seasons. At the annual scale, $C_{BB}$ emissions are found to contribute to 23% of the total (primary) contribution of $CO^{>0}$, mostly from boreal regions (BONA and BOAS), while the $C_{AN}$ contribution mainly comes from TENA, EURO, CEAS and SEAS. Note that the contribution from EURO emissions is lower than TENA because this analysis focuses on the free troposphere (above 2 km AGL, see Sect. 5.1) where the long-range transport of pollution is favoured. For subsets of stronger CO anomalies (i.e. higher percentiles), the contribution of BB emissions increase, up

to 43% for $CO^{>99}$, mainly due to an increasing influence of BONA fires. At the seasonal scale, this growing role of BB emissions is essentially observed in summer when relative $C_{BB}$ contributions increase from 42% ($CO^{>0}$) to 80% ($CO^{>99}$). On average, this represents a primary contribution of 40 ppbv for the $CO^{>99}$ anomalies. BB emissions play a marginal role during the rest of the year. A slight contribution of SEAS biomass burning is found during springtime. Bey et al. (2001) have shown that the pollution from southeast Asian fires is advected toward a large-scale convergence zone

spreading over central China and then uplifted into the free troposphere where the strong westerlies ensure a rapid transport across the Pacific Ocean. However, this SEAS contribution does not appear responsible for the strongest CO



anomalies. Actually, the whole $C_{BB}$ contribution decreases from 20 ($CO^{>0}$) to 11% ($CO^{>99}$) during that season. In winter, the primary CO is essentially anthropogenic (with BB proportions below 6%). Concerning the AN source regions, the relative importance of EURO emissions (relatively to TENA, CEAS and SEAS) increases for stronger CO anomalies, whatever the season. In other words, the AN pollution plumes that contribute to strongest CO anomalies are

mainly from local origin, in contrast with BB plumes (especially in winter, spring and fall).

### 5.5.2    North America

Results at USeast show some similarities with Germany, including a growing role of fires in the strongest CO anomalies during summer, a small influence of SEAS fires during spring and a dominant contribution of anthropogenic emissions in winter (Fig. 17). Although BONA remains the dominant source in summer (from 20% in $CO^{>0}$ to 50% in

$CO^{>99}$), fires from BOAS also exhibit a strong contribution (from 10% in $CO^{>0}$ to 25% in $CO^{>99}$). One noticeable difference with Germany is the growing importance of fires during fall (essentially from TENA), although the BB relative contribution remains moderate (from 17% in $CO^{>0}$ to 32% in $CO^{>99}$). In addition, the absolute total contributions are substantially higher than in Germany, with seasonal mean contributions in $CO^{>0}$ ($CO^{>99}$) ranging between 12 (40) ppbv in fall and 23 (60) ppbv in spring (summer). The overall picture remains the same at USlake (see

Fig. S12 in the Supplement), except that BB emissions tend to contribute more to the strongest CO anomalies, especially during spring and fall. This is particularly true during spring and fall seasons, when their contribution in $CO^{>99}$ reaches 38 and 54%, respectively, mainly due to an stronger contribution from BOAS (in spring) and BONA (in fall).

Located on the Pacific coast, the CAwest airports are mostly influenced by Asian pollution advected over the northern

Pacific by the westerlies (Fig. 18). The main sources of primary CO are BOAS fires during summer and CEAS anthropogenic emissions during the other seasons. The relative contribution of BB emissions in summer increases from 45% in $CO^{>0}$ to 92% in $CO^{>99}$. The contribution of springtime BOAS fire noticed in the strongest anomalies at USlake is not observed at CAwest. The absolute total contributions at CAwest are higher than at USeast and USlake, in particular for the strongest wintertime anomalies (when they reach 70 ppbv in $CO^{>99}$). Averaged over 2004-2012, the

Microwave Limb Sounder (MLS) observations of CO in the upper troposphere and lower stratosphere (UTLS) showed that trans-Pacific transport of CO from Asia to North America is strongest during spring and summer (Huang et al., 2016). Note that considering all points ($CO^{>0}$), the seasonal variations of the mean contribution remain moderate with values ranging between 15 ppbv during fall and 20 ppbv during spring. This amplitude is substantially lower than what is usually calculated in Eulerian global models with regionally-tagged CO emissions. For instance, at a coastal station

(elevation of 480 m) in the Washington state, Liang et al. (2004) reported Asian (from Siberia to Indonesia) CO contributions ranging between 15-20 ppbv in summer and 40-50 ppbv in spring with the GEOS-Chem model. This is due to the fact that the FLEXPART backward simulations in SOFT-IO are limited to 20 days (on purpose, in order to catch only the signature of the recent emissions, while the older pollution is expected to be well diluted after 20 days). The high springtime CO contribution given by Liang et al. (2004) results from the accumulation of primary CO during

winter/spring, which cannot be reproduced in SOFT-IO. Liang et al. (2004) reported episodic CO enhancements of 20-40 ppbv in the observations, due to trans-Pacific transport of Asian plumes, which is roughly consistent with our contributions.

### 5.5.3    Asia and India

Over Japanese airports, the contribution of fires is much lower than at CAwest, up to only 23% of $CO^{>99}$ at the annual

scale (Fig. 19). They are the dominant contributor only for the very strongest anomalies in summer (for which it reaches 60%), essentially from BOAS. The extreme anomalies are less frequent in summer than in winter/spring but some are





still observed almost every summer. Although the outflow from BOAS fires is preferentially advected over northern Japan and Sea of Okhostk (Jeong et al., 2008; Tanimoto et al., 2009), Siberian fire plumes also reach the southern parts of Japan, as observed at several urban and mountain stations in spring/summer (e.g. Kato et al., 2002; Kaneyasu et al., 2007; Ikemori et al., 2015). A moderate contribution of SEAS fires persist in spring (up to 10% in $CO^{>99}$). The

influence of EQAS fires during fall remains extremely low (as in winter). Almost no IAGOS profiles are available over Japan during the intense EQAS fires of 2015. However, even during the strong ENSO event of fall 1997 when intense fires were hitting Indonesia, airborne CO measurements in the South and East China Sea have highlighted no particular fire footprint in the upper troposphere close to Japan (Matsueda and Inoue, 1999). Whatever the anomalies subset considered, the dominant $C_{AN}$ contribution originates from CEAS at 60-90% (except in summer when it decreases to

40% in $CO^{>99}$).

At ChinaSE (Fig. 20), the $C_{BB}$ contribution increases from 30% in $CO^{>0}$ to 70% in $CO^{>99}$ at the annual scale. The main source region is SEAS during spring, followed by EQAS during fall. Previous studies already highlighted an impact of the intense Indonesian fires of 1997 at Hong-Kong (e.g. Chan et al., 2001). During winter and summer, the contribution of fires remains much lower, in particular in the most extreme events. In addition, a minor influence of NHAF fires is

observed in winter. In terms of $C_{AN}$ contribution, the main source regions are SHSA (especially in winter), CEAS and SEAS. Many similarities are found at AsiaSE (Fig. S13 in the Supplement). One difference is the much higher contribution of EQAS fires that dominates the SEAS contribution at the annual scale, in particular in the strongest CO anomalies (from 20% in $CO^{>0}$ to 60% in $CO^{>99}$). Other differences are the higher role of AsiaSE fires during winter (with a contribution reaching 40% in the $CO^{>99}$ subset) and the lower contribution from SHSA anthropogenic emissions

in winter (although strong anomalies are rare at this season).

At the SouthIndia airport cluster, the anthropogenic sources are predominant with contributions of about 80-90% at the annual scale (Fig. 21). The main source region is SEAS with some other minor contributions from CEAS, NHAF and SHSA. Only the fall fires from EQAS are found to play a role in the strongest CO anomalies of up to 40%. In summer, emissions sources from SHAF also contribute to the pollution background, but not to the strongest plumes.

### 5.5.4    Windhoek

The regional contribution at Windhoek are shown in Fig. 22. Fires play a dominant role at this airport with annual $C_{BB}$ contributions ranging from 60 ($CO^{>0}$) to 90% ($CO^{>99}$). The main source regions are SHAF and SHSA during both summer and fall, the former contributing the most to strongest CO anomalies. In winter, the fires from NHAF also show a strong contribution but strong CO anomalies are extremely sparse during that season (see Sect. 5.3). The $C_{AN}$

contributions are dominated by the SHAF source region, followed by SHSA and NHAF.

### 5.5.5    Large-scale impact of CO source regions

In this section, we summarize the long-range impact of the different (anthropogenic and biomass burning) emission source regions as seen at our airport clusters.

In terms of anthropogenic contributions, the EURO emissions contribute essentially to the pollution in Germany where

they play a predominant role in the strongest anomalies observed during winter, spring and fall. Their contribution to the Japan and North American clusters remains below a few ppbv whatever the season and the anomalies subset. The TENA anthropogenic emissions impact the airport clusters located in the eastern part of the North America (USeast, USlake) during all seasons, and can contribute substantially to the strongest CO anomalies observed in spring and, more rarely, in winter. Advected across the North Atlantic by the westerlies, this primary pollution also impact Germany but

is not found to be responsible for the strongest anomalies. However, these TENA emissions do not impact the north-western part of the continent (CAwest) that is mostly influenced by the anthropogenic pollution from CEAS and at a





lower extent from SEAS during all seasons except summer. The strong CEAS emissions also slightly contribute to the strongest anomalies at USeast and USlake (and at a lower extent in Germany).

Compared to CAwest, a similar but amplified picture is observed at the Japan cluster located directly under the anthropogenic outflow from China that highly contributes to the strongest anomalies. Japan is also impacted by the

anthropogenic emissions from SEAS. Located further south on the coast, the ChinaSE cluster is much less impacted by the CEAS anthropogenic emissions that contribute approximately the same than the anthropogenic emissions from SEAS and SHSA at the annual scale. This last region is found to contribute predominantly to the (rare) strong anomalies observed at ChinaSE. A quite similar anthropogenic contribution from these 3 regions is observed at AsiaSE, except that the SHSA contribution at AsiaSE is much lower in winter and summer. The SouthIndia cluster is essentially

influenced by the SEAS anthropogenic emissions. At Windhoek, the anthropogenic contribution is low and originates mainly from SHAF.

In terms of biomass burning emissions, the summertime BONA fires strongly impact the clusters in eastern North America and Europe, where they make a major contribution to the strong anomalies that are frequently observed. The contribution of summertime BOAS fires is also visible at these airports but much stronger on the north-western North

America (CAwest). Compared to CAwest, these BOAS fires have a lower (although still large) impact at the Japan cluster, due to its most southern location. In contrast with BONA, the BOAS fires can start as soon as spring (see Fig. 2), but the contribution from these earlier fires is only observed at USlake and Japan.

The other important source region for biomass burning is SEAS during spring. At clusters in North America and Europe, their contribution remains low and is not found to be responsible for the strongest anomalies. Note that the

contributions from SEAS anthropogenic and biomass burning emissions usually remains comparable at these airports. The SEAS fires also impact Japan but have their strongest influence at AsiaSE and overall ChinaSE. At this last cluster, the most extreme anomalies (mainly observed in spring) are largely due to these SEAS fires. At AsiaSE, the biomass burning emissions from EQAS also play a major role during the fall season (when strong anomalies are the most frequent). They are partly responsible for the strongest plumes observed at SouthIndia. At ChinaSE, these EQAS fires

highly contribute to the primary CO in fall but less frequent strong anomalies were observed during that season (relative to spring).

In summer/fall, the SHAF fires are the dominant sources of primary CO at Windhoek, followed by SHSA. The NHAF fires also contribute during winter but very few strong anomalies are observed at Windhoek during that last season.

### 5.6    Vertical distribution of SOFT-IO contributions

The climatological vertical distribution of the different anthropogenic and biomass burning contributions is shown in Fig. 23 for the entire dataset ($CO^{>0}$).

At the Germany cluster, EURO (TENA) contributions are the strongest below 6 km (10 km). The BONA contributions is the strongest in the lower free troposphere and decrease with altitude more quickly than the TENA contributions. The contributions from Asian source regions (CEAS, SEAS) reach their maximum higher in altitude, roughly between 6 and

12 km. In particular, the SEAS contribution peaks at about 10 km. At the clusters in North America, the TENA emissions mainly impact the lower altitudes while strongest contributions from CEAS and SEAS are found higher in the troposphere, between 4 and 12 km (with a maximum between 6 and 10 km). Interestingly, a small contribution from EAQS fires is highlighted at CAwest in the upper troposphere, above 11 km. This EQAS pollution at such a high altitude may be explained by the frequent presence of deep convective systems over the maritime continent (Hong et al.,

2008), which allows a rapid uplift of the pollution in the higher troposphere where long-range transport is favoured.



At the Japan cluster, the BONA contribution is strongest in the lower part of the free troposphere while the CEAS contribution is important in the entire free troposphere. As in North America, the SEAS and EQAS contributions are maximum in the higher part of the free troposphere. Based on GEOS-Chem simulations, Bey et al. (2001) showed that CO from SEAS is mainly exported in the free troposphere and not so much in the boundary layer in contrast with the

CEAS CO export that occurs in both layers. This difference is due to the relative latitudinal position of these two types of emissions, the anthropogenic emissions being located at more northerly latitudes than the biomass burning emissions. Bey et al. (2001) also indicated that deep convection mainly occurs in southeast Asia during spring season. This is consistent with the lower SEAS contribution observed here in the lower free troposphere at Japan airports.

Apart from the monsoon summer season (and especially in winter), due to the presence of the Siberian High and the

Aleutian Low in Pacific Ocean, north-easterly winds at the surface bring continental polluted air masses to the south-eastern part of Asia (Wu and Wang, 2002). Contributions from SEAS and CEAS source regions at ChinaSE and AsiaSE clusters thus peak in the lower troposphere. During fall season, an additional contribution from EQAS is found through the entire free troposphere with a maximum at 10 km. During the summer monsoon period, convective activity induces a very different vertical distributions with substantial contributions through the entire free troposphere with a maximum

in the higher troposphere.

The main characteristic at SouthIndia is the seasonal variability of the anthropogenic SEAS contribution that is maximum in the lower troposphere during all seasons except in summer when it clearly maximizes higher in the troposphere. The convective uplift of pollution to the upper troposphere is a well known phenomena associated with the Asian summer monsoon and confirmed by numerous airborne and satellite observations (e.g. Kar et al., 2004; Jiang et

al., 2007; Barret et al., 2016).

At Windhoek, the vertical distribution of the CO contributions during the fire seasons (JJA and SON) is maximum in the lower free troposphere, mainly due to the contribution from SHAF. The contribution from SHSA extends higher in altitude and peaks at around 7 km. Over South America, the biomass burning pollution plumes can be uplifted at high altitudes with deep convective systems and then transported by the westerlies near 25°S and around transient

anticyclones toward southern Africa (Stohl, 2004). A secondary maximum is found above 8 km in summer with contributions from SHAF biomass burning emissions and NHAF anthropogenic emissions. At this season, the ITCZ is located high in northern latitude (about 15°N) and the pollution emitted in this region can be transported in the Hadley cell before reaching the high altitudes above Windhoek.

## 6    Summary and conclusion

In the framework of IAGOS, vertical profiles of tropospheric CO have been routinely measured at worldwide international airports since 2002. In these profiles, strongly polluted CO plumes are frequently encountered by the IAGOS aircraft. This paper has investigated the role of biomass burning in these plumes and the associated long-range transport. Results are analysed at 9 clusters of nearby airports in different parts of the world, namely Europe, North America, Asia and Southern Africa. Considering the large IAV of biomass burning emissions in many source regions

and the episodic nature of long-range transport mechanisms, an important aspect of this work is the long time period considered (2002-2017, i.e. 16 years) during which about 30,000 CO profiles were analysed. Compared to spatially and temporally limited research campaigns, this allows to catch a more representative picture of the role of fires.
In order to help the interpretation of the IAGOS measurements, we first gave a brief overview of several important features of the CO biomass burning emissions (from the GFAS inventory), including their spatio-temporal variability,

latitudinal distribution, IAV and trends. Biomass burning emissions exhibit a strong regional, seasonal and inter-annual



variability. Inter-regional and inter-annual differences of emissions typically exceed one order of magnitude. Intrinsically linked to the meteorological conditions and biomass availability, they are characterized by a strong seasonal variability, with maximum emissions occurring during dry seasons. Although the time period is likely still too short to provide robust conclusions, some statistically significant trends were highlighted, including a decrease of CO
biomass burning emissions at the global scale (-1.7±1.0% yr$^{-1}$) and in Southern Hemisphere South America (-5.1±3.8% yr$^{-1}$) maybe due to a reduced deforestation over the recent years.

We provided an altitude-dependent distribution of CO mixing ratios based on the entire IAGOS dataset (about 125 million observations) in order to give the most general view of the CO levels typically encountered in the troposphere. Concerning the vertical distribution of the extreme CO mixing ratios registered by IAGOS over 2002-2017, the 99[th]
(99.9[th]) percentile decreases with altitude from 750 (1619) ppbv below 1 km to 162 (229) ppbv above 12 km of altitude. The overview of all individual CO vertical profiles at the different airport clusters highlight frequent but irregular strong CO plumes in the free troposphere at most locations. In order to investigate the role of biomass burning relative to anthropogenic emissions, we simulated the recent primary CO contribution from both types of sources along all IAGOS trajectories with the recently developed SOFT-IO tool. Reproducing (usually vertically thin) pollution plumes traveling
at the global scale with Eulerian chemistry-transport models remains a challenging task, notably due to the dilution of the plumes in the coarse grid. SOFT-IO addresses this problem by coupling FLEXPART retroplume simulations (over 20 days) with CO emission inventories, which allows to estimate the (recent) contribution of primary CO emissions with additional information on emission types (anthropogenic or biomass burning) and source regions. Although many uncertainty sources persist (e.g. emissions, transport), SOFT-IO is able to provide valuable information.
In this study, anomalies at each airport cluster are defined as departures from the seasonally-averaged climatological vertical profile. This study focuses on the free troposphere (here roughly defined as the part of troposphere above 2 km AGL) where long-rage transport is favoured. The variability of CO mixing ratios around the climatological profile greatly differs from one region to the other. Among the different airport clusters, the strongest CO anomalies were found at Asian clusters where the 99[th] percentile of the CO anomalies ranges between 117 and 151 ppbv, and the lowest
(48 ppbv) in Germany. An analysis of the seasonal distribution of the highest CO anomalies in the free troposphere exhibits a large seasonal variability at all locations. Except over Japan and South India where anthropogenic CO dominates, the SOFT-IO results demonstrated the growing role of biomass burning sources in the strongest CO anomalies observed at all airport clusters in the free troposphere. The vertical distribution of the frequency of occurrence of these CO plumes greatly differs from one region to the other, with for instance more frequent strong
anomalies in the lower free troposphere in Asia, Germany and at Windhoek (in Namibia) at the annual scale, in contrast with North America where they tend to be more equally distributed throughout the troposphere, although some differences exist at the seasonal scale.

We investigated the long-range transport of these plumes by analysing the contributions from the different source regions in the world. Over Germany, strong anomalies are observed in winter and spring, due to anthropogenic
emissions from Europe and United States, with a small contribution from Asia. During summertime, the strongest anomalies are mostly due to boreal North America fires. These fires are also clearly responsible for the strongest anomalies observed at airports located in northeastern North America, in addition to a substantial contribution from boreal Asia fires. The impact of these last fires is strong and clearly dominant during boreal summer over the IAGOS airports located on the north-west coast. At these airports, the anthropogenic emissions from central-east Asia also
strongly contribute to the anomalies observed during the other seasons. Over Japan, the strongest anomalies are recorded more frequently in winter and spring, mostly due to the anthropogenic emissions from central-east Asia, although biomass burning from southeast and boreal Asia also substantially contribute to the springtime anomalies. In



southern China, the strongest anomalies are mostly observed during spring season due to the biomass burning emissions in south-east Asia. In the southern part of the Indochina peninsula, the strongest anomalies are distributed all along the year except during the Asian summer monsoon. The spring (fall) anomalies are mostly caused by biomass burning from south-east Asia (equatorial Asia), while wintertime anomalies are due to biomass burning from south-east Asia in

combination with anthropogenic emissions from several regions. In south India, anomalies are also observed during all non-monsoon seasons and are essentially due to the anthropogenic emissions from southeast Asia, except in fall when fires from equatorial Asia are found to contribute up to 40% to the strongest anomalies. At Windhoek, all strongest anomalies are observed in fall and summer and caused essentially by the intense biomass burning emissions over southern hemisphere Africa and South America. The vertical distribution of these regional contributions also reveals

useful information on the long-range transport from these different source regions.

In this paper, the role of biomass burning was investigated at many different locations from a climatological point of view. It provides both qualitative and quantitative information for interpreting the highly variable CO vertical profiles in these regions of interest. However, dedicated studies in specific regions are obviously required to get a more detailed understanding about how these fires impact the chemical composition of the troposphere. This study made extensive use

of Lagrangian modelling, which may offer some interesting opportunities for comparisons with Eulerian modelling. Note also that an on-going complementary study based on the IAGOS measurements obtained during the cruise phase will complete our understanding of these issues in the upper troposphere and lowermost troposphere. Another rich although complex perspective would be to investigate the ozone formation in these plumes along their long-range transport and maybe to identify different signatures depending on the source regions (due to different initial chemical

composition of the plume and/or different environment).

**Data availability**

No new measurements were made for this review article. The IAGOS data are available on http://www.iagos.fr or directly via AERIS web site http://www.aeris-data.fr. The SOFT-IO products will be made available through the

IAGOS central database (http://iagos.sedoo.fr/#L4Place) and are part of the ancillary products (https://doi.org/10.25326/3, Sauvage et al., 2017a, 2017b).

**Author contributions (temporary)**

Contributed to conception and design : HP

Contribution to acquisition of data : HP, VT, BS, GA, RB, DB, J-MC, PN
Contributed to analysis and interpretation of data : HP, BS, VT, MP
Drafted the article : HP
Revised the article :

**Competing interests**

The authors have no competing interests to declare.

**Acknowledgement**

The authors acknowledge the strong support of the European Commission, Airbus, and the Airlines (Lufthansa, Air-France, Austrian, Air Namibia, Cathay Pacific, Iberia and China Airlines so far) who carry the MOZAIC or IAGOS

equipment and perform the maintenance since 1994. In its last 10 years of operation, MOZAIC has been funded by





INSU-CNRS (France), Météo-France, Université Paul Sabatier (Toulouse, France) and Research Center Jülich (FZJ, Jülich, Germany). IAGOS has been additionally funded by the EU projects IAGOS-DS and IAGOS-ERI. The MOZAIC-IAGOS database is supported by AERIS (CNES and INSU-CNRS). Data are also available via AERIS web site www.aeris-data.fr.

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





**Table 1 : Description of airport clusters (the number of profiles is reported into brackets).**

| Airport cluster | List of airports |
|---|---|
| Germany (14,197) | Frankfurt (11,573), Munich (1,902), Dusseldorf (722) |
| USeast (2,480) | New York (969), Philadelphia (754), Boston (496), Washington (261) |
| USlake (1,630) | Toronto (865), Chicago (573), Detroit (192) |
| CAwest (1,430) | Vancouver (774), Portland (426), Calgary (230) |
| Japan (2,733) | Tokyo (1,256), Nagoya (1,135), Osaka (342) |
| ChinaSE (2,429) | Taipei (1,806), Hong-Kong (441), Guangzhou (114), Xiamen (68) |
| AsiaSE (1,603) | Bangkok (1,426), Ho Chi Minh City (177) |
| SouthIndia (1,114) | Hyderabad (581), Madras (498), Mumbai (35) |
| Windhoek (1,937) | Windhoek (1,937) |

**Table 2: Mean CO emissions from vegetation fires and interannual variability (IAV).** Mean emissions are
5 calculated over the period 2002-2017. The IAV are given at the annual and seasonal scale.

| Region | Description | Mean CO emissions [TgCO yr$^{-1}$] (contribution [%]) | Annual IAV [%] | Minimum and maximum seasonal IAV [%] |
|---|---|---|---|---|
| WORLD | World | 361 (100%) | 13 | 12 (DJF) – 28 (SON) |
| BONA | Boreal North America | 19 (5%) | 38 | 29 (SON) – 84 (MAM) |
| TENA | Temperate North America | 7 (2%) | 26 | 26 (DJF) – 45 (JJA) |
| CEAM | Central America | 6 (2%) | 27 | 19 (DJF) – 43 (JJA/SON) |
| NHSA | Northern Hemisphere South America | 5 (1%) | 27 | 25 (SON) – 42 (MAM) |
| SHSA | Southern Hemisphere South America | 51 (14%) | 38 | 25 (MAM) – 49 (JJA) |
| EURO | Europe | 1 (<1%) | 39 | 27 (MAM) – 80 (SON) |
| MIDE | Middle East | 2 (<1%) | 46 | 39 (SON) – 67 (JJA) |
| NHAF | Northern Hemisphere Africa | 56 (15%) | 11 | 13 (DJF) – 42 (JJA) |
| SHAF | Southern Hemisphere Africa | 71 (20%) | 8 | 9 (JJA) – 24 (MAM) |
| BOAS | Boreal Asia | 47 (13%) | 49 | 66 (JJA) – 107 (SON) |
| CEAS | Central Asia | 12 (3%) | 26 | 33 (DJF) – 62 (JJA) |
| SEAS | South East Asia | 24 (7%) | 21 | 17 (JJA) – 25 (DJF) |
| EQAS | Equatorial Asia | 38 (11%) | 90 | 57 (JJA) – 128 (SON) |
| AUST | Australia | 21 (6%) | 42 | 30 (MAM) – 58 (SON) |





**Table 3: Seasonal linear trends of CO emissions from vegetation fires (in % yr$^{-1}$) over 2002-2017.** Uncertainties are given at a 95% confidence level (NS means non-significant). The significance (here defined as the trend normalized by the uncertainty at a 95% confidence level) is given into brackets. The mean annual CO emissions (in TgCO yr$^{-1}$) are remained into square brackets in the first column.

| Region | DJF | MAM | JJA | SON | ANN |
|---|---|---|---|---|---|
| WORLD [361] | -1.8±1.2 (1.6) | NS | NS | -2.8±2.1 (1.3) | -1.7±1.0 (1.7) |
| BONA [19] | NS | NS | NS | NS | NS |
| TENA [7] | NS | NS | NS | NS | NS |
| CEAM [6] | NS | NS | NS | NS | NS |
| NHSA [5] | NS | NS | 3.5±3.4 (1.0) | NS | NS |
| SHSA [51] | NS | -3.6±2.4 (1.5) | -6.1±4.9 (1.2) | -5.0±3.9 (1.3) | -5.1±3.8 (1.3) |
| EURO [1] | -4.7±4.3 (1.1) | NS | NS | NS | NS |
| MIDE [2] | 5.1±4.1 (1.3) | 7.9±7.6 (1.04) | 5.3±4.9 (1.1) | 6.9±5.3 (1.3) | 6.2±5.0 (1.2) |
| NHAF [56] | -2.1±1.2 (1.7) | NS | NS | NS | NS |
| SHAF [71] | NS | NS | NS | NS | NS |
| BOAS [47] | NS | NS | NS | NS | NS |
| CEAS [12] | NS | NS | NS | -8.3±4.6 (1.8) | -4.0±2.5 (1.6) |
| SEAS [24] | NS | NS | 2.2±1.8 (1.2) | 3.4±1.5 (2.3) | NS |
| EQAS [38] | NS | NS | -6.5±6.1 (1.1) | NS | NS |
| AUST [42] | NS | NS | NS | NS | NS |

**Table 4 : Distribution of the CO anomalies (in ppbv) at the different airport clusters.** Anomalies are defined as the departures from the seasonally-averaged climatological vertical profile (see text).

| Airport cluster | Percentile of CO mixing ratios anomalies (ppbv) | | | | | | | | | |
|---|---|---|---|---|---|---|---|---|---|---|
| | 0[th] | 5[th] | 25[th] | 50[th] | 75[th] | 80[th] | 90[th] | 95[th] | 98[th] | 99[th] |
| Germany | -72 | -26 | -12 | -2 | 9 | 12 | 21 | 29 | 39 | 48 |
| USeast | -70 | -29 | -14 | -3 | 10 | 14 | 24 | 34 | 48 | 62 |
| USlake | -73 | -29 | -14 | -3 | 10 | 14 | 24 | 34 | 49 | 64 |
| CAwest | -68 | -29 | -14 | -2 | 10 | 13 | 23 | 33 | 50 | 67 |
| Japan | -108 | -47 | -28 | -12 | 12 | 21 | 50 | 81 | 120 | 151 |
| ChinaSE | -156 | -45 | -21 | -6 | 13 | 18 | 38 | 62 | 104 | 140 |
| AsiaSE | -142 | -38 | -17 | -3 | 12 | 17 | 32 | 50 | 81 | 117 |
| SouthIndia | -93 | -30 | -14 | -3 | 9 | 12 | 23 | 35 | 53 | 69 |
| Windhoek | -88 | -33 | -15 | -4 | 10 | 14 | 25 | 39 | 65 | 96 |



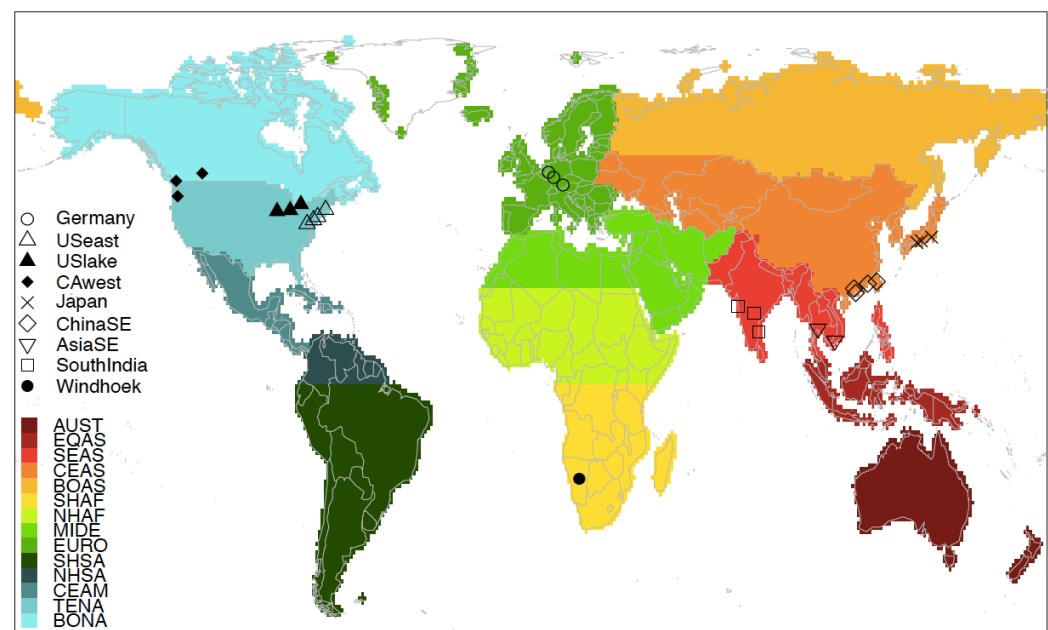

**Figure 1: Geographical regions (from GFED) and airport clusters**.

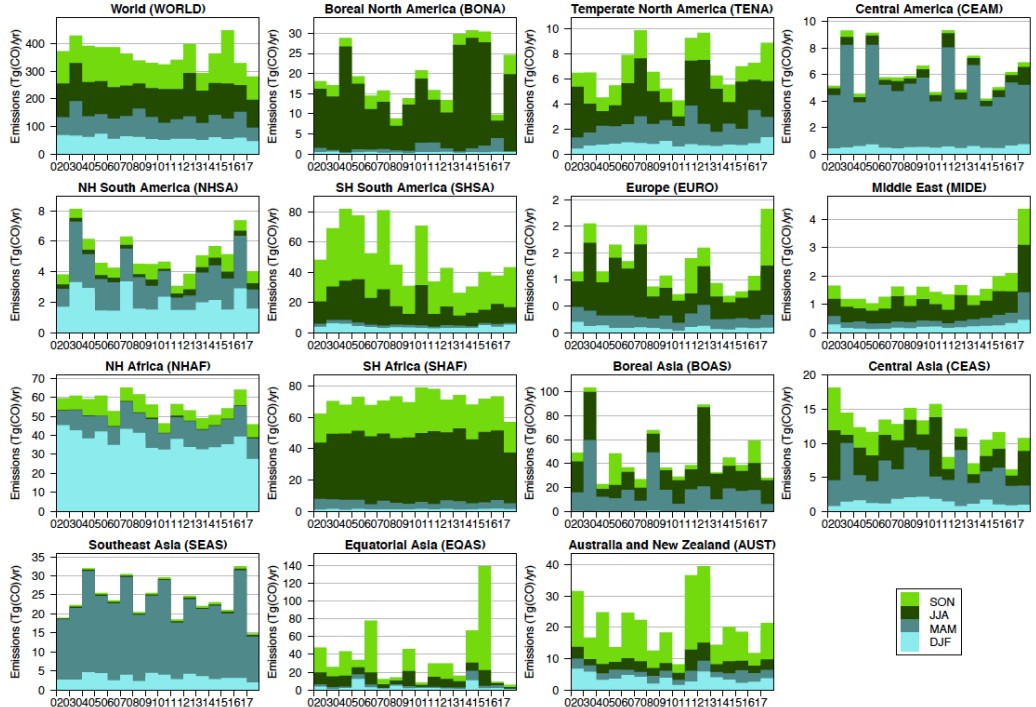

**Figure 2: Cumulated seasonal GFAS biomass burning CO emissions at the global scale and in the 14 continental**

**regions**. Emissions are from GFASv1.0 in 2002 and from GFASv1.2. over the 2003-2017 period.





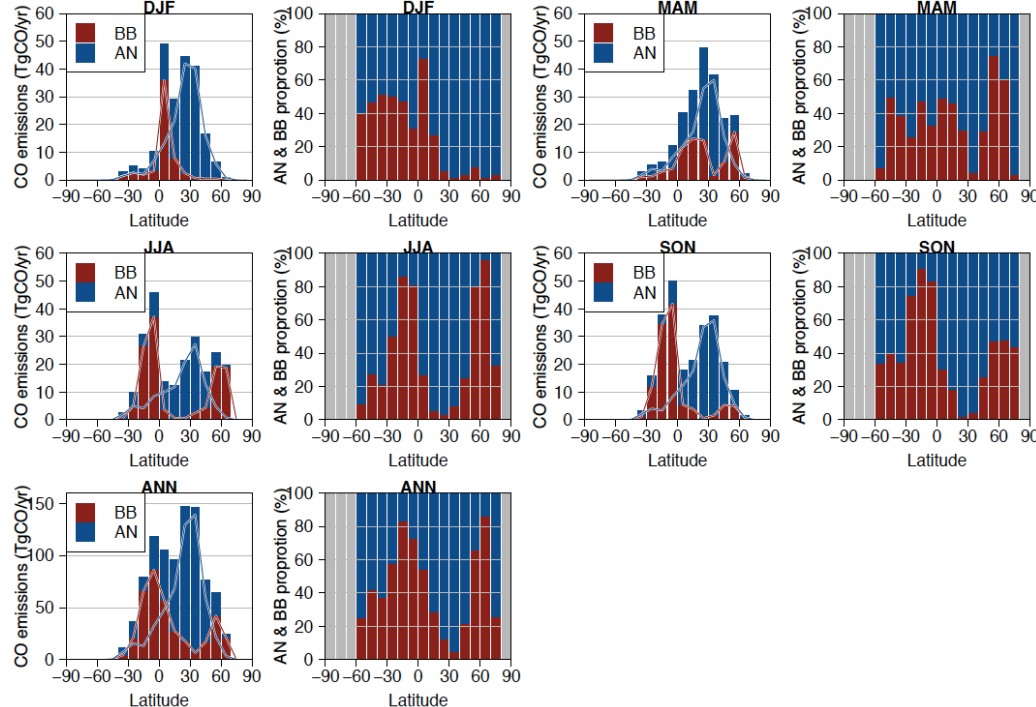

**Figure 3: Latitudinal distribution of anthropogenic (AN) and biomass burning (BB) CO emissions.** Emissions are given at the annual scale, in absolute (1st and 3rd column) and relative (2nd and 4th column). In the plots of absolute emissions, the CO emissions of BB and AN emissions are shown by lines, while the bars indicate the cumulate of both sources.

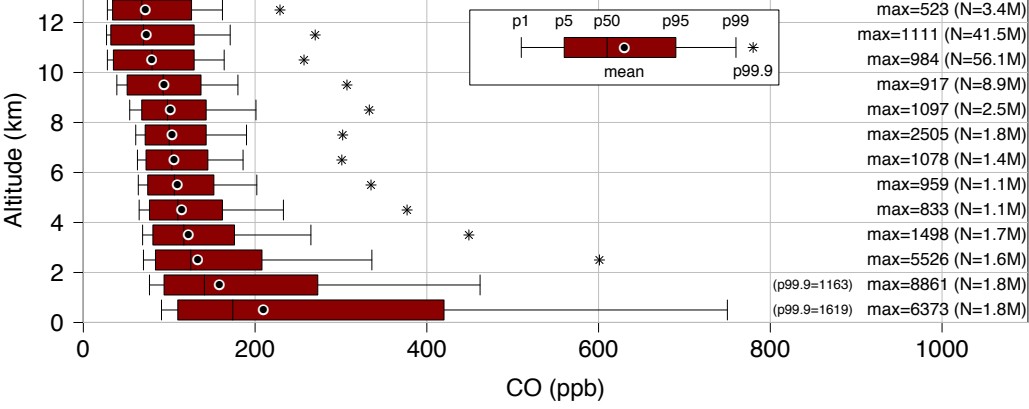

**Figure 4: Distribution of CO mixing ratios measured by MOZAIC-IAGOS aircraft during the 2002-2017 period.** Results are shown per 1 km–width layer, without any discrimination between troposphere and stratosphere (N gives the number of points in millions, pX in the legend indicates the Xth percentile of the distribution).





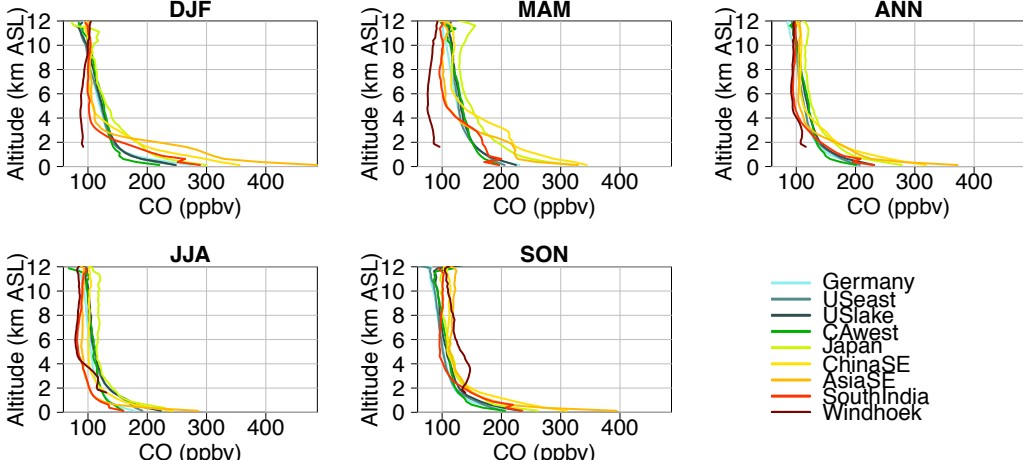

**Figure 5: Mean seasonal profiles of CO mixing ratios at the different clusters.**

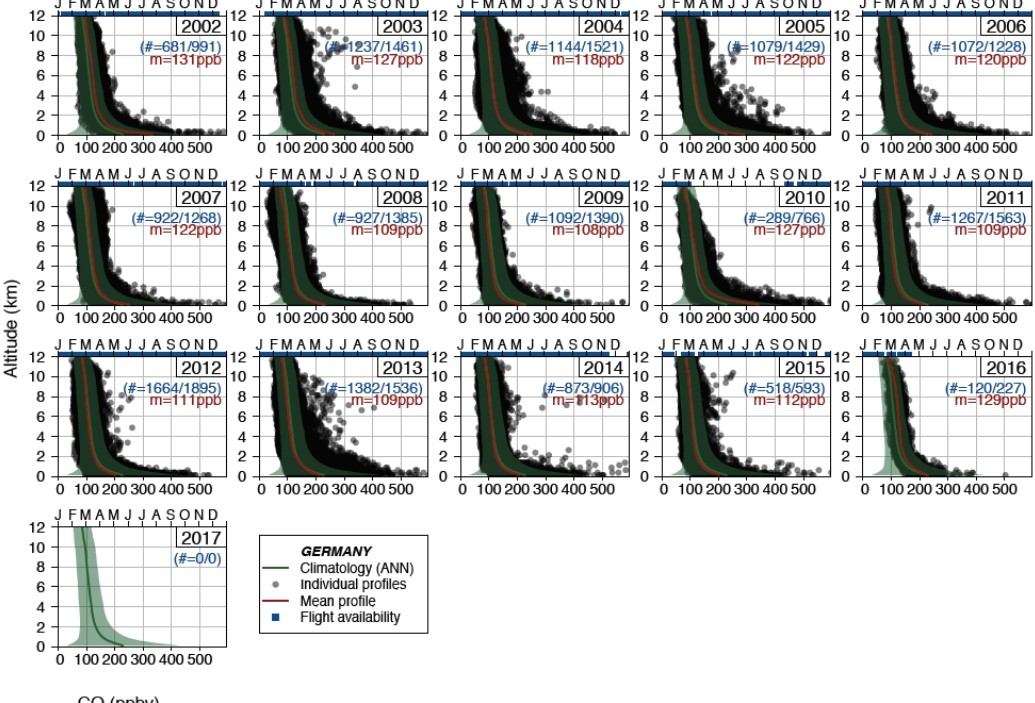

**Figure 6: Overview of CO mixing ratio profiles at Germany.** All individual CO profiles sampled between 2002 and 2017 are shown with black points (a transparency is added to better highlight the density of points), and the corresponding average profile in red. The mean climatological vertical profile over the period 2002-2017 is shown with a green line (the green area corresponds to ±2σ). The plot also shows in blue the number of vertical profiles and their distribution along the year (a blue bar on the time axis above graphs indicates that a flight is available on that day; this time axis does not correspond to the abscissa of the plot). Vertical profiles in 2017 are not yet in a validated status at Germany airports.



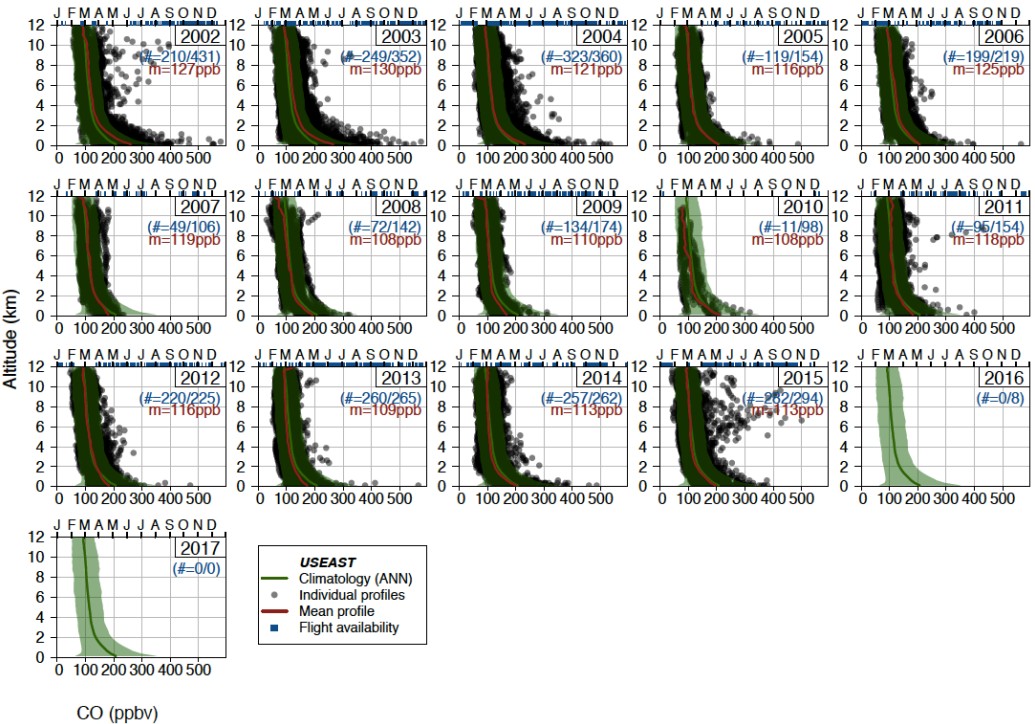

**Figure 7: Same as Fig. 6 at USeast cluster.**

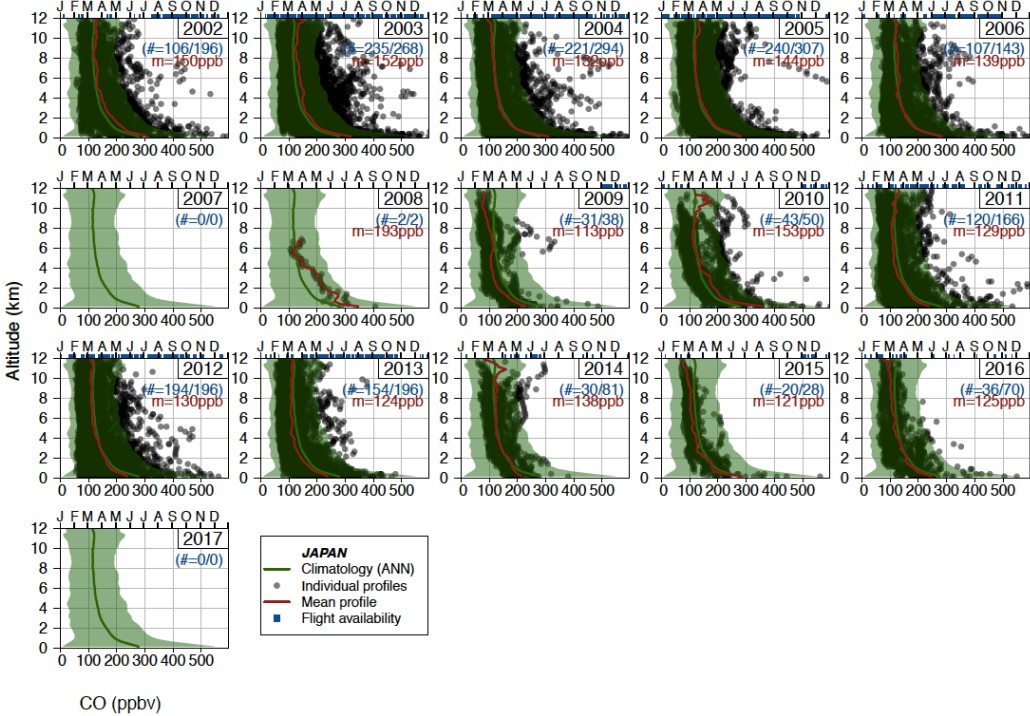

**Figure 8: Same as Fig. 6 at Japan cluster.**




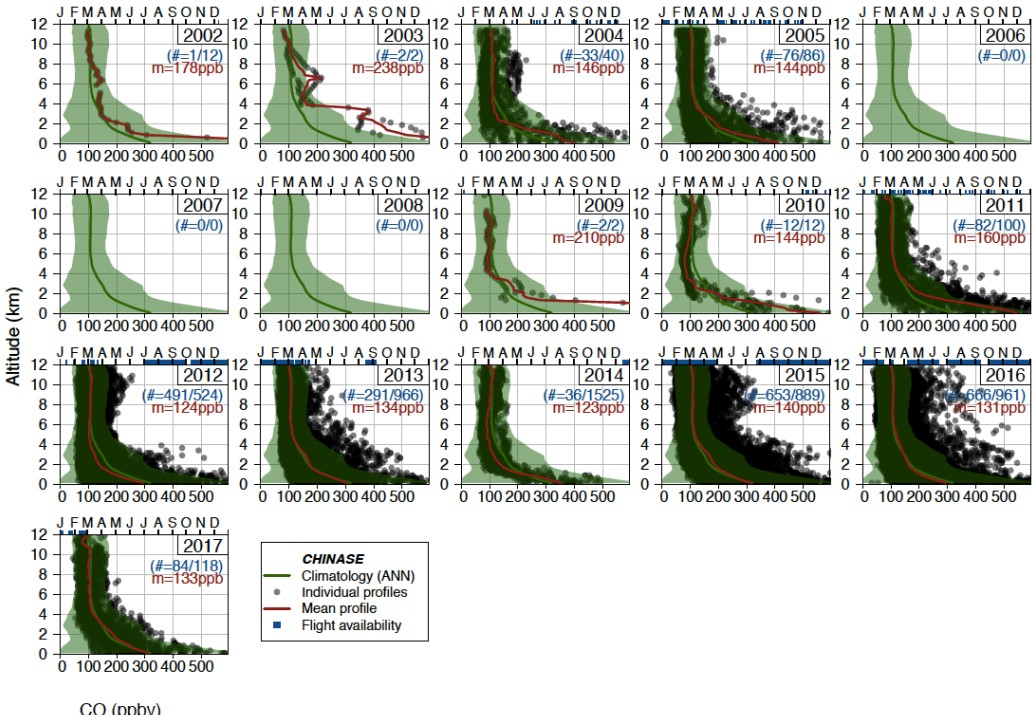

Figure 9: Same as Fig. 6 at ChinaSE cluster.

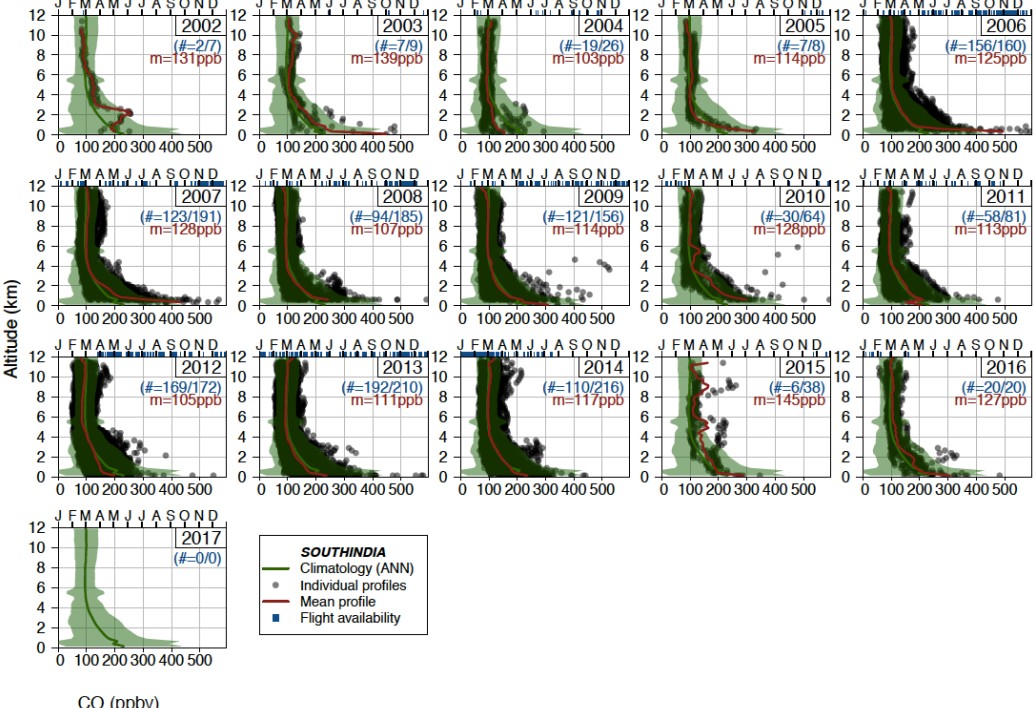

Figure 10: Same as Fig. 6 at SouthIndia cluster.



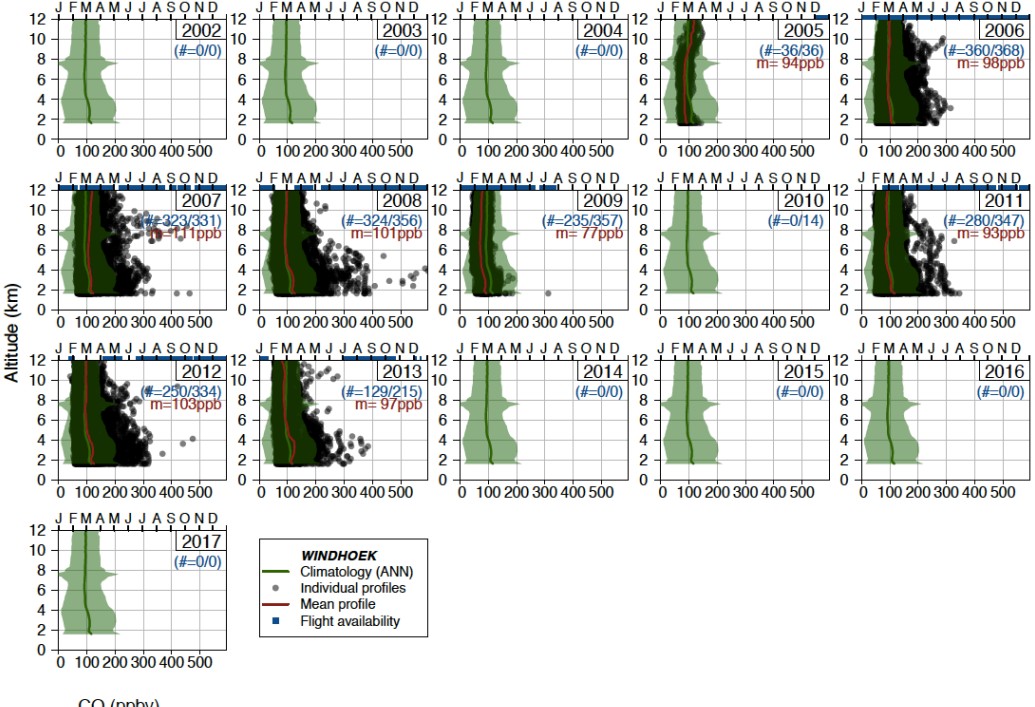

**Figure 11 : Same as Fig. 6 at Windhoek cluster.**

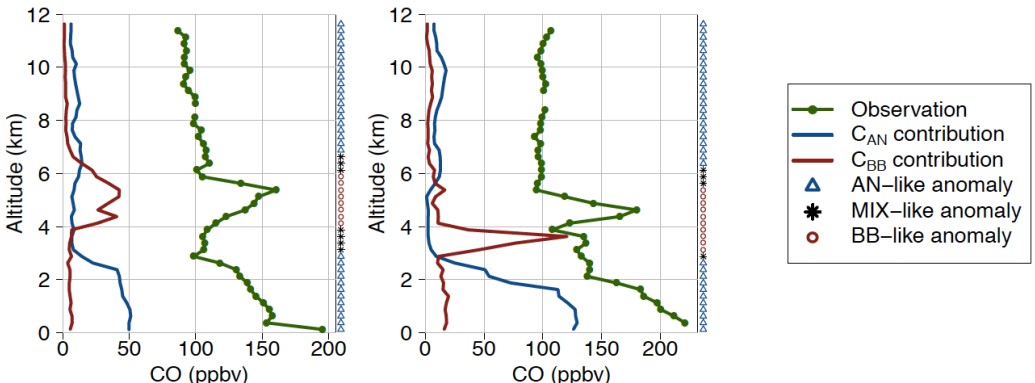

**Figure 12: Vertical profiles of CO at New York airport on August 28th and 30th 2004.** Observed CO mixing ratios are green, $C_{BB}$ and $C_{AN}$ contributions simulated with SOFT-IO are in red and blue, respectively. The class assigned to the observed anomalies based on the $C_{BB}/C_{AN}$ ratio (see text) is indicated on the right side of each plot (AN-like anomalies with blue triangles, MIX-like anomalies with black stars, BB-like anomalies with red circles).





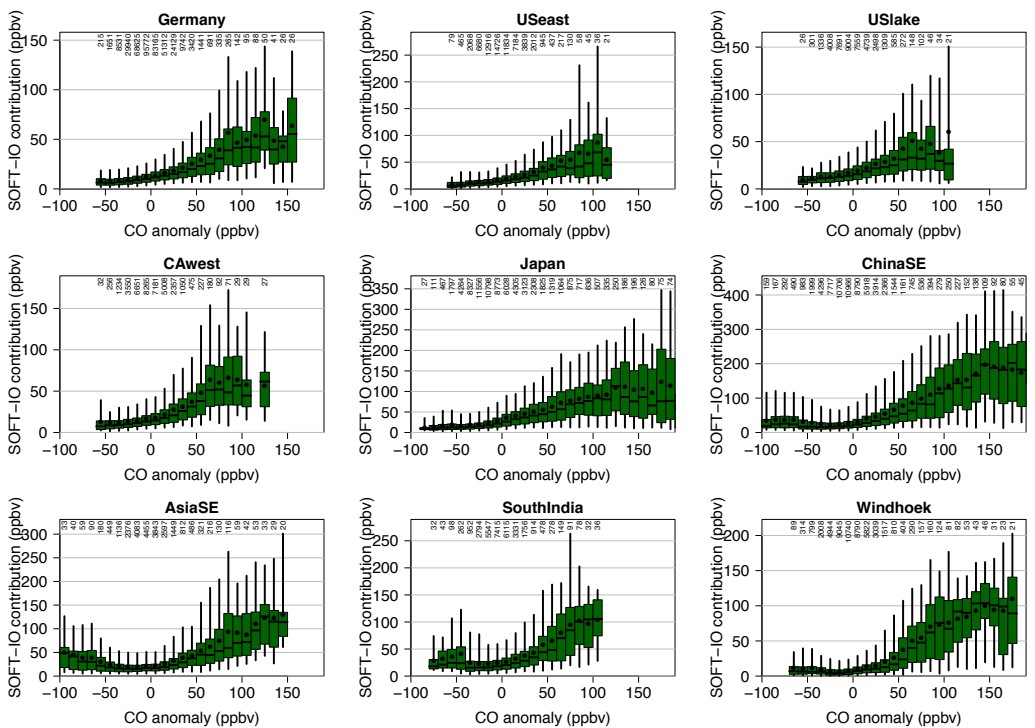

5    **Figure 13: SOFT-IO total (C$_{AN+BB}$) contributions against observed CO anomalies.** The box-a-whisker plot shows the 5th, 25th, 50th, 75th and 95th percentiles of the contributions, the black dot indicates the mean contribution. The number of points included is reported on the top of each panel. No distribution is plotted when this number is below 20.





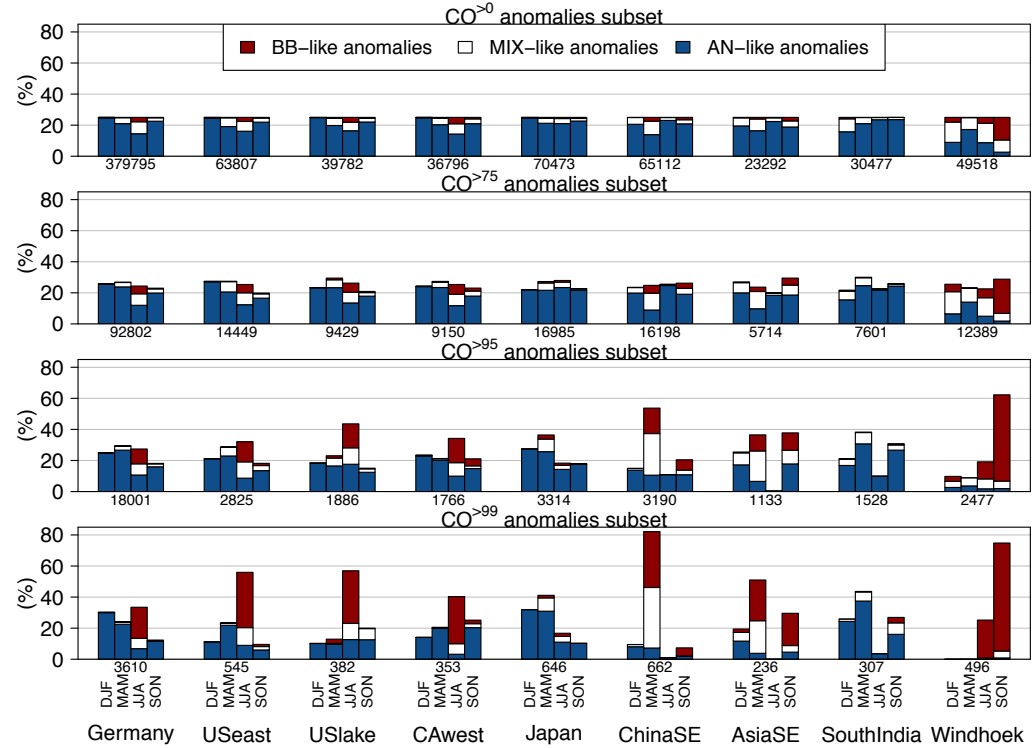

**Figure 14: Seasonal distribution of the BB, MIX and AN anomalies at the different airports.** Several subsets of CO anomalies are shown : $CO^{>0}$ (first row), $CO^{>75}$ (second row), $CO^{>95}$ (third row) and $CO^{>99}$ (last row). The frequency of occurrence is weighted by the number of available data during each season. The number of CO anomalies is also indicated below each bar plot.






**Figure 15 : Seasonal and annual vertical distribution of the frequency of occurrence of the CO$^{>75}$, CO$^{>95}$ and CO$^{>99}$ anomaly subsets.** At all altitudes, the frequency of occurrence of CO anomalies is weighted by the number of available data during each season (in order to allow comparisons between the different seasons).

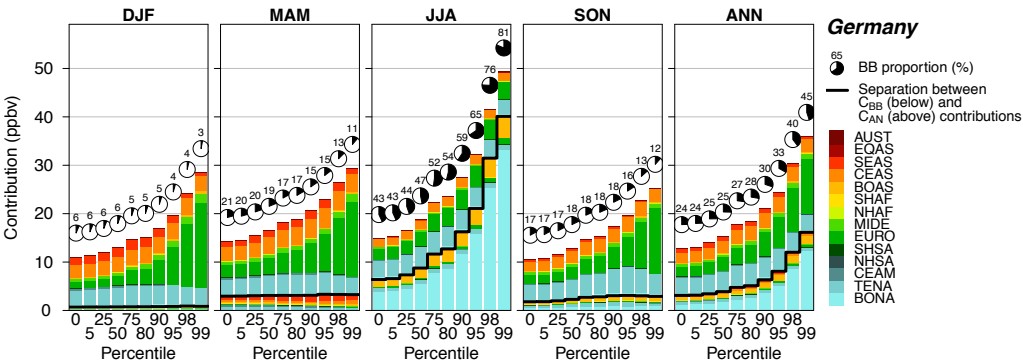

**Figure 16: Mean total (C$_{AN+BB}$) contributions to CO anomalies at the Germany airport cluster.** Results are shown for all seasons and different anomalies subsets (designated by the corresponding percentile). The geographic origin of both types of CO emissions is indicated by the colours. The dark line separates the C$_{BB}$ (below) and C$_{AN}$ (above) contributions. The relative contribution of BB in the total (AN+BB) primary contribution is indicated with a pie chart and the corresponding figure.

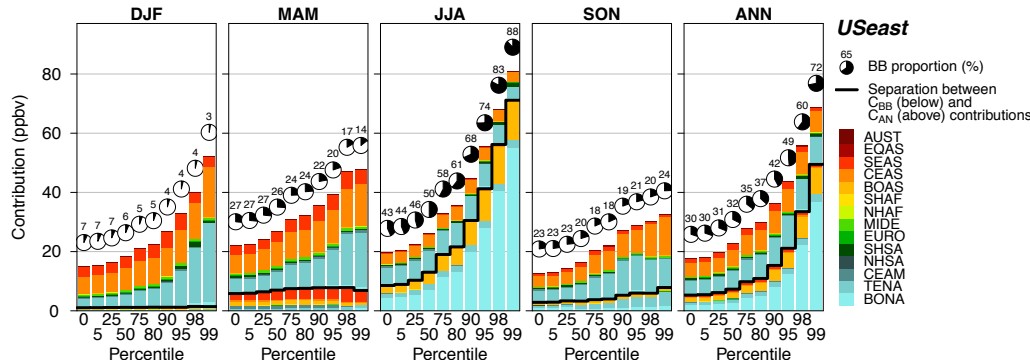

15    **Figure 17: Same as Fig. 16 for USeast cluster.**





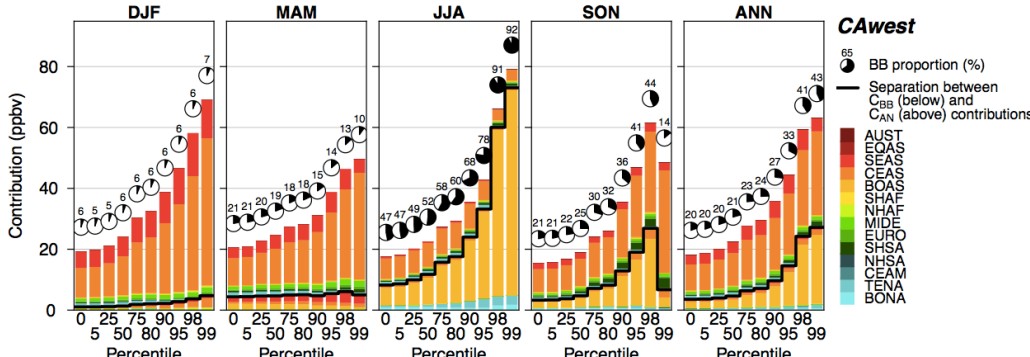

**Figure 18: Same as Fig. 16 for CAwest cluster.** Note that the much lower BB contribution during fall in $CO^{>99}$ is due to a low number of points.

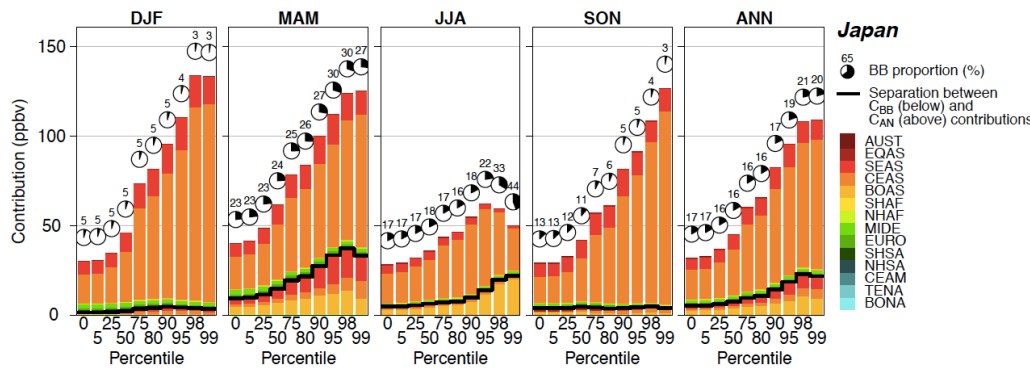

5    **Figure 19: Same as Fig. 16 for Japan cluster.**

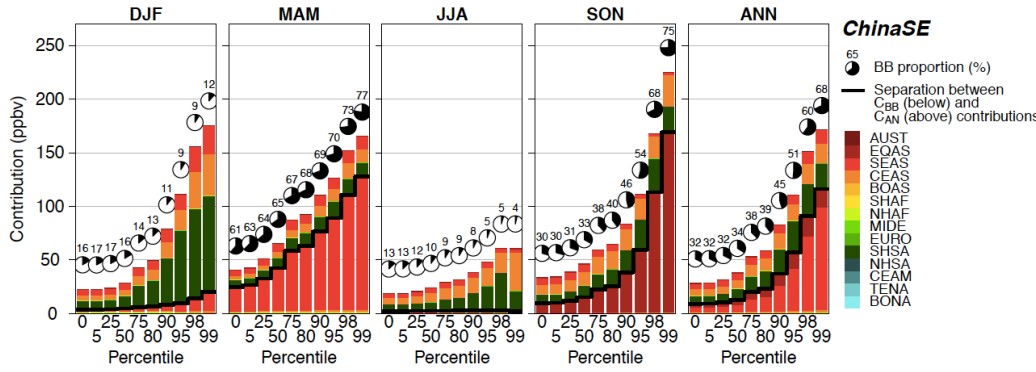

**Figure 20: Same as Fig. 16 for ChinaSE cluster.**





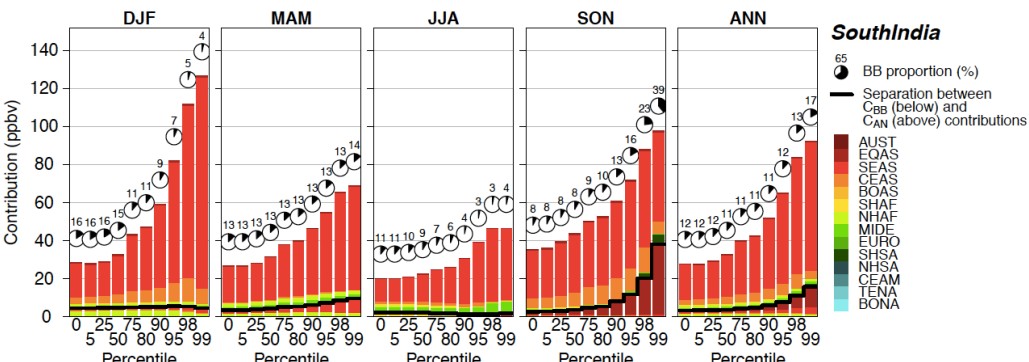

5      **Figure 21: Same as Fig. 16 for SouthIndia cluster.**

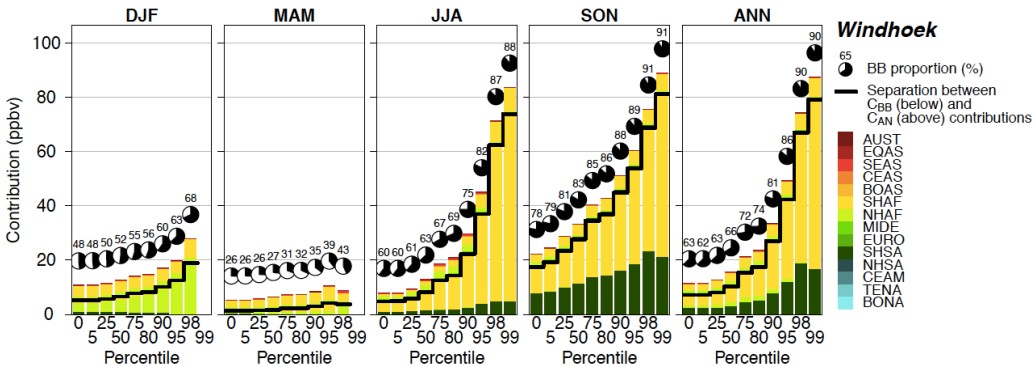

10     **Figure 22: Same as Fig. 16 for Windhoek cluster.**







**Figure 23: Climatological vertical distribution of the SOFT-IO total ($C_{AN+BB}$) contributions, averaged over the all IAGOS vertical profiles ($CO^{>0}$).** The colours indicate the source regions. The total biomass burning contributions is shown with a black line (anthropogenic contributions are thus on the right of this black line).