# Peer review of "Tropospheric CO vertical profiles measured by IAGOS aircraft in 2002-2017 and the role of biomass burning"

_Atmospheric Chemistry and Physics, 2018_

## Referee Comment (RC1) · Anonymous Referee #1 · 27 Aug 2018

**General:**
The paper presents a very comprehensive analysis of long-term CO observations performed within the IAGOS project. In particular the observed profiles, mainly in the vicinity of the airports, are extensively discussed. The observed (positive) anomalies from the climatological CO profiles are traced back to their sources by using the backward trajectories technique (FLEXPART) and by including emission inventories containing anthropogenic and biomass burning data. The results show, in a very impressive way, the importance of the biomass burning for the understanding of the observed CO profiles. Because the data set is so large, statistical approach is necessary which was excellently performed in this paper. Consequently, a very robust picture is drown show-

ing how the anthropogenic and biomass burning sources massively contribute to the observed anomalies. The paper is well-written (also little bit too long, see below). The figures are excellent (also some small improvement are still possible, see below). The presented analysis is very clean and covers the issue from all different angles. Thus, I would like to recommend this paper for publishing by ACP with only some minor points listed below.

**Minor comments:**

- Title
  Because your main results are related to the biomass burning maybe: "The role of biomass burning as derived from...."

- P3 L4-15
  Please write out the abbreviations like MOZAIC, IAGOS or FLEXPART if they are used first time in the manuscript.

- P4 L1
  What do you mean with "fully validated"? It sounds very technical

- P4 L15-20
  Please add which type of met. data is used in SOFT-IO, ERA-Interim or ECMWF Analysis or even something very different (like MERRA-2)

- P5 L29
  "carbon fuel content" - what do you mean in context of the biomass burning

- Figure 2 and 3
  Both matrix-figures can be optimized by removing some redundant x-axis and y-axis captions. In this way the sub-panels become larger and easier to read.

- P6
  For me section 3.3 is much too long. It needs to much time to reach the most interesting part of the paper starting with section 4.

- P7 L5-9
  How do average vertically the profiles shown in Figure 4

- P7 L15
  "1, 100" - here is something wrong with the notation

- Title of section 4.2
  Maybe "Seasonality of climatological...."

- P7, L25
  ...are 10-30 ppbv higher than...

- Figure 6
  The profile availability is very difficult to read. Also the blue and red numbers are not sufficiently explained

- P9 L36
  even if explained before ("will focus on the strongest positive CO anomalies") I would recommend to write: "...represents the whole positive anomalies dataset"

- Figure 12
  one y-axis caption "Altitude" would be enough

- Figure 13
  I would replace the y-axis label "SOFT-IO contribution..." by "$C_{AN+BB}$ contribution..." and remove the redundant x- and y-labels in this matrix figure

- P11-12, section 5.4, Figure 14
  For me this figure and the related explanation takes too much space in your
paper. There are no clear conclusions from this figure. Also it slightly disturbs the "dramatic line" of your paper because it belongs roughly to the previous part i.e. around Fig. 12 (i.e. where the quantities $C^{>p}$ are introduced). Maybe you can shift and shorten this part.

---

## Referee Comment (RC2) · Anonymous Referee #2 · 18 Sep 2018

The article "Tropospheric CO vertical profiles measured by IAGOS aircraft in 2002–2017 and the role of biomass burning" is very well written and clear; its scientific significance is certainly high as it seems to be the first effort on this scale to quantify the biomass burning vs anthropogenic origin of CO plumes. The authors show a very good grasp of the IAGOS dataset, and how to use it for extreme events; they are careful not to draw general conclusion when the number of events/observations is too small. The paper is well structured and very informative.

In short, I have no major comment and I think this paper can be published nearly as is. The only few questions remarks that I have are:

[Figure]

- In Figures 6 to 11, perhaps density plots (ie scatterplots with a different color code depending on the density) could show better the information with such a number of observations.

- Figure 12 is intended as an example to show how SOFT-IO provides the anthropogenic and biomass-burning contribution of CO concentration. The sum of the two is however very much below the observed profile, even at a low altitude. Surely the difference cannot be entirely explained as secondary CO or CO that was emitted more than 20 days ago (especially close to the surface)? If possible, an explanation would be welcome.

- The legends of Figures 15 and 23 are not on the same page as the Figures themselves.

---

## Author Comment (AC1) · 30 Oct 2018

**Answers to the first reviewer**

*General:*
*The paper presents a very comprehensive analysis of long-term CO observations performed within the IAGOS project. In particular the observed profiles, mainly in the vicinity of the airports, are extensively discussed. The observed (positive) anomalies from the climatological CO profiles are traced back to their sources by using the backward trajectories technique (FLEXPART) and by including emission inventories containing anthropogenic and biomass burning data. The results show, in a very impressive way, the importance of the biomass burning for the understanding of the observed CO profiles. Because the data set is so large, statistical approach is necessary which was excellently performed in this paper. Consequently, a very robust picture is drown showing how the anthropogenic and biomass burning sources massively contribute to the observed anomalies. The paper is well-written (also little bit too long, see below). The figures are excellent (also some small improvement are still possible, see below). The presented analysis is very clean and covers the issue from all different angles. Thus, I would like to recommend this paper for publishing by ACP with only some minor points listed below.*

We thank the reviewer for his/her positive appreciation and comments. In the revised manuscript, we took into account all his/her suggestions. In the following, the comments are in blue and the answers in black.

*Minor comments:*
*Title : Because your main results are related to the biomass burning maybe: "The role of biomass burning as derived from...."*
We followed the suggestion of the reviewer, the title is now : « The role of biomass burning as derived from the tropospheric CO vertical profiles measured by IAGOS aircraft in 2002-2017 »

*P3 L4-15 : Please write out the abbreviations like MOZAIC, IAGOS or FLEXPART if they are used first time in the manuscript.*
Done, page 3, line 2 : « Frequent profiles with high vertical resolution are essential for better characterizing biomass burning plumes and their transport. In the framework of the MOZAIC (Measurement of Ozone by Airbus In-service aircraft) program and its successor the IAGOS (In-service Aircraft for a Global Observing System) European Research Infrastructure [...] »

*P4 L1 : What do you mean with "fully validated"? It sounds very technical*
We replaced "fully validated" by "calibrated".

*P4 L15-20 : Please add which type of met. data is used in SOFT-IO, ERA-Interim or ECMWF Analysis or even something very different (like MERRA-2)*
We added : "The meteorological fields are taken from the ECMWF analysis and forecast. "

*P5 L29 : "carbon fuel content" - what do you mean in context of the biomass burning*

It simply means how much carbon is found in the biomass. We modified the text as follows page 5 line 29 : « In boreal regions, several factors drive the intensity of biomass burning emissions, including weather, carbon fuel content (quantity of carbon in the biomass) and topography. »

*Figure 2 and 3: Both matrix-figures can be optimized by removing some redundant x-axis and y-axis captions. In this way the sub-panels become larger and easier to read.*
We modified these two figures.

*P6 : For me section 3.3 is much too long. It needs to much time to reach the most interesting part of the paper starting with section 4.*
To our opinion, this section 3.3 is not particularly long. As the CO GFAS emissions are one of the main input data in the study, we think they deserve an appropriate description. We still reduced the second part of this section 3.3 (page 6, lines 24-39) : "[...] The most noticeable and strongest annual trend (-5.1±3.8% yr$^{-1}$ or -2.6±2.0 TgCO yr$^{-1}$) is observed in SHSA where CO emissions are decreasing during all seasons except winter (mostly in summer and fall). This is consistent with Chen et al. (2013) that highlighted an increase of the number of active fires over 2001-2005 followed by a slight decrease (and large IAV), notably due to a substantial reduction of deforestation in Brazil over the 2000s (Reddington et al., 2015). Small significant decreases are observed in some other regions, including NHAF during the fire season and CEAS during fall. A strong but weakly significant decrease is also observed during summertime in EQAS (-6.5±6.1% yr$^{-1}$ or -0.6±0.5 TgCO yr$^{-1}$). Due to surprisingly higher emissions in 2017 (a factor 2-3 higher than over the period 2002-2016), the MIDE shows significant positive trends during all seasons but CO emissions in this region are very low. The strong emissions in 2017 are probably artificially caused by an out-of-date mask for filtering of oil and gas flaring hotspots in the GFAS system, which would not cover the more recent activities in this region.  In most other regions, no significant trends are found. "

*P7 L5-9: How do average vertically the profiles shown in Figure 4*
We are not sure to understand what is not clear here. In Figure 4, we simply calculated the different metrics (average, percentiles 5th, 25th, 50th, 75th, 95th and 99th) considering all the IAGOS observations in the different 1-km intervals of altitude.

*P7 L15 : "1, 100" - here is something wrong with the notation*
We corrected in "1100 ppbv".

*Title of section 4.2 : Maybe "Seasonality of climatological...."*
In this short section, we are briefly describing the climatological profiles but the discussion does not focus on the seasonality (which is investigated in more details in the rest of the paper), thus we think that the title should not be modified.

*P7, L25 : ...are 10-30 ppbv higher than...*
Correction applied.

*Figure 6 : The profile availability is very difficult to read. Also the blue and red numbers are*
*not sufficiently explained*

In Figure 6, the reader is not supposed to be able to see easily all the dates when IAGOS profiles are available. The profile availability is provided in order to give him a quick and overall view of the availability of the profiles. We have worked a lot on this figure in order to give all the information in a condensed format and we do think that its current form should not be modified. In the final paper, this figure will in higher resolution which will allow the reader to zoom if necessary (although we think that it is not necessary to catch our message).

Concerning the blue and red numbers, we added : "The numbers in blue indicate the number of IAGOS profiles with available CO observations and the total number of IAGOS profiles during the considered period. The number in red is the average CO mixing ratio over the entire mean profile"

*P9 L36 : even if explained before ("will focus on the strongest positive CO anomalies") I would recommend to write: "...represents the whole positive anomalies dataset"*

No, there is a misunderstanding here : the $CO^{>0}$ dataset includes all the CO observations above the $0^{th}$ percentile (i.e. the minimum), in other words the whole (positive and negative) anomalies dataset, and not only the whole positive anomalies dataset. This is clearly explained with some examples in Sect. 5.1.

*Figure 12 : one y-axis caption "Altitude" would be enough*

Correction applied.

*Figure 13 : I would replace the y-axis label "SOFT-IO contribution..." by "$C_{AN+BB}$ contribution..."*
*and remove the redundant x- and y-labels in this matrix figure*

Correction applied.

*P11-12, section 5.4, Figure 14 : For me this figure and the related explanation takes too much space in your paper. There are no clear conclusions from this figure. Also it slightly disturbs the "dramatic line" of your paper because it belongs roughly to the previous part i.e. around Fig. 12 (i.e. where the quantities $C_{>p}$ are introduced). Maybe you can shift and shorten this part.*

We assumed that the reviewer is here talking about Fig. 15 and not Fig. 14 (since Fig. 14 is one of the most important figure of the paper). Although we somehow understand the comment of the referee, we consider that it is important to describe the vertical distribution of the anomalies since only the IAGOS dataset can allow such analysis. It notably shows that the vertical distribution of the strongest CO anomalies ($CO^{>99}$) can vary substantially from one airport to the other and can differ strongly compared to the less intense anomalies ($CO^{>75}$). We agree with the referee that the paper is quite long, but considering the scale of the analysis (15 years of data at many airport clusters in different regions), we think that its size remains appropriate. In addition, as it has only 378 words (less than 5% of the paper), removing or shortening this section would not reduce substantially the length of the paper.

We just applied some minor modifications to reduce slightly the discussion (page 12, lines 5-18) : "At the annual scale, the $CO^{>75}$ anomalies are quite equally distributed in the free troposphere at most airport clusters, with low to moderate differences are observed at the seasonal scale. In comparison, larger inter-seasonal and inter-regional differences are found for the $CO^{>95}$ and $CO^{>99}$ subsets. At the Germany cluster, the strongest anomalies tend to be more frequent in the lower part of the free troposphere, except in spring and summer when anomalies are found higher in altitude. At USeast and USlake, the strongest anomalies are more equally distributed in the troposphere although the frequency of occurrence drops above 10-11 km. Different results are observed at CAwest where the strong anomalies are the most frequent above 4-5 km in winter, spring and fall and in the lower troposphere in summer. At Japan airports, frequent strong anomalies are observed in the upper troposphere (above 10 km) in spring. At ChinaSE and AsiaSE, the strongest anomalies are clearly more frequent in the lower free troposphere in spring and extend higher in the troposphere during fall. At SouthIndia, frequent anomalies are also in the lower free troposphere during all non-monsoon seasons, with a secondary maximum of frequency in the upper troposphere. At Windhoek, the strongest anomalies are restricted to the lower free troposphere during the burning season, except during fall when frequent strong anomalies are also observed higher in altitude, up to 10-11 km."

---

## Author Comment (AC2) · 30 Oct 2018

**Answers to the second reviewer**

*The article "Tropospheric CO vertical profiles measured by IAGOS aircraft in 2002–2017 and the role of biomass burning" is very well written and clear; its scientific significance is certainly high as it seems to be the first effort on this scale to quantify the biomass burning vs anthropogenic origin of CO plumes. The authors show a very good grasp of the IAGOS dataset, and how to use it for extreme events; they are careful not to draw general conclusion when the number of events/observations is too small. The paper is well structured and very informative. In short, I have no major comment and I think this paper can be published nearly as is.*

We thank the reviewer for his/her positive review and his/her comments. In the following, the comments are in blue and the answers in black.

*The only few questions remarks that I have are:*
*In Figures 6 to 11, perhaps density plots (i.e. scatterplots with a different color code depending on the density) could show better the information with such a number of observations.*

As explained to the other reviewer, we do think that this figure is the most readable in its current form. The idea of this figure is to give a brief and overall view of the IAGOS profiles at a given airport cluster. To our opinion, adding colors to illustrate the density of points would strongly complicate the figure for a poor additional level of understanding. We thus think it should remains in its current form. Note that we are not much interested here in the density of points within the ±2σ around the climatological profile (i.e. the green area), but we want to shed light on the stronger CO mixing ratios where the transparency is directly useful.

*Figure 12 is intended as an example to show how SOFT-IO provides the anthropogenic and biomass-burning contribution of CO concentration. The sum of the two is however very much below the observed profile, even at a low altitude. Surely the difference cannot be entirely explained as secondary CO or CO that was emitted more than 20 days ago (especially close to the surface)? If possible, an explanation would be welcome.*

As explained in the paper, SOFT-IO does not simulate the CO background but only the contributions from recent (less than 20 days) emissions. In Figure 12, the background (i.e. the CO profile minus the $C_{AN+BB}$ profile) is roughly 100 ppbv of CO and does not change strongly with altitude. This corresponds to the order of magnitude of what is expected for the sum of secondary CO and primary CO older than 20 days. The formation of secondary CO is rather slow and thus the secondary CO is not expected to be concentrated close to the surface. Similarly, after 20 days and considering the intermediate chemical lifetime of CO, the old primary CO is also expected to be quite equally distributed in the troposphere. Therefore, to our opinion, the results shown in Fig. 12 do not appear unrealistic.

*The legends of Figures 15 and 23 are not on the same page as the Figures themselves.*

Sorry for the inconvenience but this is only a Word draft for discussion. The problem will of course be solved in the final publication.